# Multi-omics microsampling for the profiling of lifestyle-associated changes in health

Xiaotao Shen [1,2,5], Ryan Kellogg[1,2,5], Daniel J. Panyard [1,2,5], Nasim Bararpour[1,2,5], Kevin Erazo Castillo[1,2], Brittany Lee-McMullen[1,2], Alireza Delfarah[1,2], Jessalyn Ubellacker[1], Sara Ahadi[1,2], Yael Rosenberg-Hasson[3], Ariel Ganz [1,2], Kévin Contrepois[1,2], Basil Michael[1,2], Ian Simms[1,2], Chuchu Wang [4], Daniel Hornburg[1,2] & Michael P. Snyder [1,2] ✉

Current healthcare practices are reactive and use limited physiological and clinical information, often collected months or years apart. Moreover, the discovery and profiling of blood biomarkers in clinical and research settings are constrained by geographical barriers, the cost and inconvenience of in-clinic venepuncture, low sampling frequency and the low depth of molecular measurements. Here we describe a strategy for the frequent capture and analysis of thousands of metabolites, lipids, cytokines and proteins in 10 µl of blood alongside physiological information from wearable sensors. We show the advantages of such frequent and dense multi-omics microsampling in two applications: the assessment of the reactions to a complex mixture of dietary interventions, to discover individualized inflammatory and metabolic responses; and deep individualized profiling, to reveal large-scale molecular fluctuations as well as thousands of molecular relationships associated with intra-day physiological variations (in heart rate, for example) and with the levels of clinical biomarkers (specifically, glucose and cortisol) and of physical activity. Combining wearables and multi-omics microsampling for frequent and scalable omics may facilitate dynamic health profiling and biomarker discovery.

Multi-omics technologies enable the quantification of thousands of molecules and can provide new insights into the molecular landscape of health and disease[1,2]. Despite major advances in omics technologies, the upstream sample collection and processing still requires travel to a clinic, access to a phlebotomist and physical and emotional discomfort. These current sample-collection strategies do not meet the desired flexibility and non-invasiveness to conduct comprehensive longitudinal profiling independent of access to a clinic. Furthermore, the high sample volume needed (often 10–50 ml of venous blood) prohibits frequent collections, which precludes high-resolution analysis of dynamic metabolic and biological processes that occur on the scale of minutes

or hours. Finally, high sample collection and processing costs can be prohibitive for performing large studies in remote environments.

Previous studies have investigated dried blood spot (DBS) sampling[3–6] and volumetric absorptive microsampling (VAMS)[7–9] for metabolite and protein analyses[10]. In principle, DBS allows individuals to collect a blood drop sample at home and return the sample by mail at room temperature. However, DBS sampling is often irreproducible since volumetric amounts can vary considerably, and, so far, the number of analytes analysed from DBS has generally been modest[11].

In this Article, to circumvent these challenges, we devised a streamlined multi-omics profiling system that uses finger prick blood drop

[1]Department of Genetics, Stanford University School of Medicine, Stanford, CA, USA. [2]Stanford Center for Genomics and Personalized Medicine, Stanford, CA, USA. [3]Human Immune Monitoring Center, Microbiology and Immunology, Stanford University Medical Center, Stanford, CA, USA. [4]Howard Hughes Medical Institute, Stanford University, Stanford, CA, USA. [5]These authors contributed equally: Xiaotao Shen, Ryan Kellogg, Daniel J. Panyard, Nasim Bararpour. ✉e-mail: mpsnyder@stanford.edu

collection, minimizes pain and enables sampling frequencies on the timescale of minutes without needing clinic access. Our method collects fixed 10 µl volumes and, following extraction, enables the simultaneous analysis of proteins, metabolites, lipids and targeted cytokines/hormones from a single sample enabling broad analyte profiling. In two proof-of-principle studies, we first demonstrate the profiling of a dynamic response to ingestion of a mixed meal shake and discover high heterogeneity in individual metabolic and immune responses, and second, we perform high-resolution profiling of an individual over 1 week enabling the identification and quantification of thousands of molecular changes and associations across 'omes' at a personal level. Our approach is scalable, enabling high-frequency molecular profiling for broad utility in research and clinical studies.

## Results

### Overview of the multi-omics microsampling approach

The blood microsampling and multi-omics data acquisition workflow are shown in Fig. 1a. After testing numerous methods, we settled on collecting 10 µl blood microsamples using a Mitra device, a solid matrix that collects fixed blood volumes. We tested a wide variety of extraction conditions and further developed a method for efficiently extracting proteins, a broad range of lipids, and metabolites from a single microsample using biphasic extraction with methyl *tert*-butyl ether (MTBE). This extraction procedure yields an organic phase containing hydrophobic metabolites and lipids, an aqueous phase containing hydrophilic metabolites and a methanol-precipitated protein pellet processed for proteomics data acquisition. Using a separate microsample, we performed an aqueous extraction for performing multiplexed immunoassays on the Luminex platform (Methods). Omics datasets were then processed, annotated and curated for detailed omics analysis.

To evaluate the microsampling method, we first examined the stability of proteins, metabolites and lipids in microsamples under multiple conditions, including testing storage duration and temperature (Fig. 1b and Extended Data Fig. 1a). We then compared microsampling with conventional intravenous sampling methods (Fig. 1b). Finally, two pilot case studies were performed to demonstrate how microsampling can capture important health and biological perturbations in a lifestyle context (Fig. 1b).

### Protein, metabolite and lipid stability in microsamples in multiple conditions

We first evaluated the stability of proteins, metabolites and lipids in the blood microsamples (Supplementary Fig. 1). In brief, blood samples were collected from two participants using the 10 µl Mitra devices. A total of 36 microsamples were collected from each participant, with the microsamples stored in duplicate at three temperatures (4, 25 and 37 °C) and for five durations at each temperature (3, 6, 24, 72 and 120 h) before storage at −80 °C until analysis. An additional set of samples was immediately stored at −80 °C. Proteomics, metabolomics and lipidomics data were acquired from the microsamples (Methods). After quality control (QC), imputation and annotation of the data, there were 66 proteomics samples with 128 proteins, 71 metabolomics samples with 1,461 annotated features and 72 lipidomics samples with 776 lipids (Supplementary Dataset 1). Each omics dataset was assessed individually to examine analyte stability concerning storage duration, storage temperature and the interaction of storage duration and temperature. The stability metrics assessed were (1) the average coefficient of variation (CV) across both participants' samples (estimated using the formula for log-scale data[12]), (2) the presence of significant effects of storage conditions on analyte level using a linear regression analysis (excluding the baseline samples that were not stored at any temperature) and (3) relative importance measures (partial $R^2$ and the Lindeman, Merenda and Gold measure, LMG1; Methods).

The results revealed that, overall, the majority of analytes were quite stable to storage duration, temperature and the interaction effect (Fig. 1c,d). Proteins were the most stable (CV range 0.149–1.728, median 0.397) with few, that is, three (2.3%), eight (6.3%) and six (4.7%), associated with storage duration, temperature and the interaction effect, respectively. Metabolites were less stable (CV range 0.054–54.328, median 0.378) with 194 (13.3%), 389 (26.6%) and 193 (13.2%) associated with storage duration, temperature and the interaction effect, respectively. Finally, lipids were the least stable (CV range 0.088–2.218, median 0.335), with 150 (19.3%), 513 (66.1%) and 172 (22.1%) associated with storage duration, temperature and the interaction effect, respectively. The relative importance models gave similar results. Thus, most analytes can be reliably measured using remote sampling, and the less stable ones can be identified and potentially measured using correction models.

### Comparison between microsample and intravenous plasma sample

We next examined the similarity between the molecular profiles derived from microsamples of whole blood compared with venepuncture plasma. Blood samples were collected from 34 participants using both microsampling and conventional intravenous blood draws (Supplementary Fig. 1a and Methods), and metabolomics and lipidomics data were acquired from each participant (Supplementary Dataset 2). The median intensity of every feature in the 34 participants was calculated separately in the two datasets, microsampling and intravenous plasma collection samples, and compared via correlation graphs (Fig. 1e). Interestingly, the results of the microsampling and intravenous collection methods were quite similar in that the Spearman correlations were 0.81 ($P < 0.001$) and 0.94 ($P < 0.001$) for 642 metabolites and 616 lipids, respectively. Metabolites and lipids that were not well correlated (Spearman correlation < 0.5) were enriched for amino acids and triglycerides (TAGs), respectively (Supplementary Fig. 1b,c). However, most classes of molecules were very similar between the microsampling and venous blood draw, including most of the amino acids, carbohydrates, free fatty acids (FFAs), TAGs, diglycerides, phosphatidylcholines (PCs) and other molecules.

### Case studies

As a demonstration of the power of microsampling, we performed two case studies while participants were in their native environments. The first was to examine the effect of drinking a complex mixture on metabolic profiles. The second was to perform very dense '24/7' profiling (98 microsamples) across a period of just longer than 7 days.

### Case study 1: metabolic phenotyping responses to Ensure shake consumption

Individuals can differ markedly in their metabolic response to food on the basis of their epigenome, microbiome, metabolome and other factors[13–16], yet the heterogeneity of this response is not well understood or fully established. Determining these differences at an individual level is important to optimize diet and lifestyle changes for personalized health, weight reduction and/or management of the metabolic disease. Biomarkers are typically measured at a single timepoint because of the difficulty of collecting high-frequency blood samples using a conventional blood sampling approach, but the rapid and dynamic nature of metabolism in response to food intake requires higher resolution. To follow the diversity of metabolic responses to complex dietary mixtures, we measured the multi-omics responses to a defined mix of carbohydrates, lipids, proteins and micronutrients. We analysed metabolomics, lipidomics, cytokines and hormones in 28 participants with diverse backgrounds (Fig. 2a and Supplementary Fig. 2a) and developed six metabolic responses metrics: (1) carbohydrate, (2) lipid, (3) amino acid (protein), (4) insulin secretion, (5) FFA (related to insulin sensitivity) and (6) immune (cytokines).

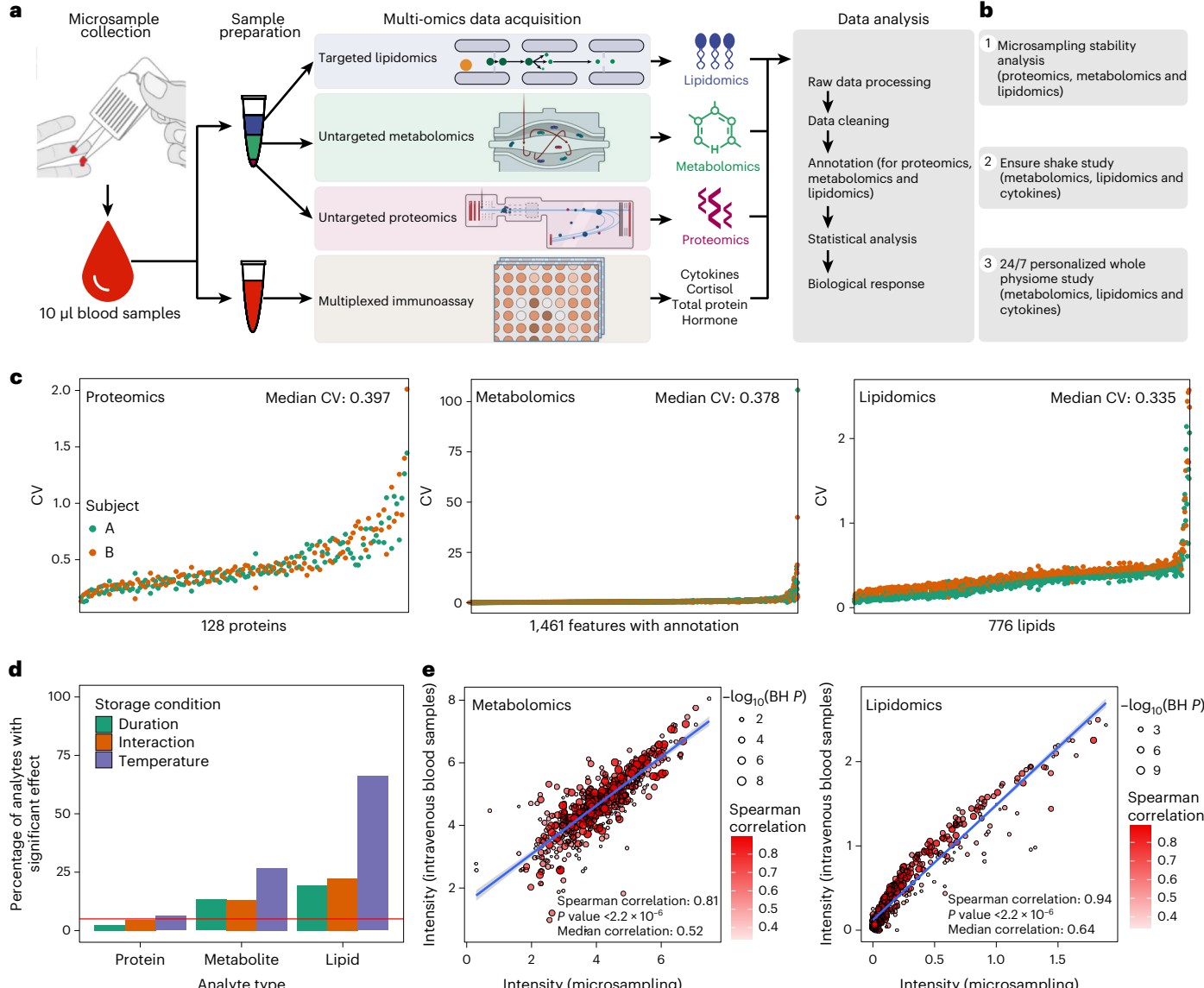

**Fig. 1 | Overview of the microsampling multi-omics workflow and stability analysis. a**, The samples were collected using microsampling devices, and then multi-omics data (proteomics, metabolomics, lipidomics, cytokine and so on) were acquired. **b**, Outline of the primary microsampling analyses. **c**, The coefficient of variation (CV) distribution for proteins, metabolites and lipids across all the samples in the stability analysis. **d**, The percentage of analytes is significantly affected by storage duration, temperature and interactions (linear regression). The red line shows the expected proportion of nominally significant results at the alpha level of 5% ($P = 0.05$). **e**, The Spearman correlations between microsamples and intravenous blood samples ($n = 34$) for metabolites and lipids, respectively.

Thirty-two participants were mailed a kit containing microsampling Mitra devices, an Ensure shake and careful instructions for microsampling sample collection. Each participant collected one microsample (defined as 0 min), consumed the Ensure shake and collected additional blood microsamples at 30, 60, 120 and 240 min after consumption (Fig. 2a). Participants returned their microsamples by overnight mail on the same day of microsample collection. The microsamples were used for multi-omics data acquisition, namely, metabolomics, lipidomics and cytokines/hormones. Four subjects without metabolomics data were removed from the dataset (Methods and Fig. 2b). After data cleaning, curation and annotation, 768 analytes were detected from the microsamples, including 560 metabolites, 155 lipids and 54 cytokines/hormones for each of the 28 participants at each of the five timepoints (a total of 140 data points) (Fig. 2b and Supplementary Dataset 3).

## Clustering of altered molecules

We first determined whether the microsampled multi-omics data reflected the consumption of the Ensure shake. For each timepoint post consumption, the Wilcoxon rank test was used to define the significantly dysregulated molecules compared with timepoint 0 (baseline). Interestingly, the majority of significantly increased metabolites and lipids peaked at approximately 60 min and 120 min, respectively, and then approached baseline levels by 240 min (Fig. 2c). These results indicate that many molecules substantially responded to Ensure shake in the blood, and the response kinetics differed on the basis of the classes of molecules.

To quantify the molecules that shifted their levels upon Ensure shake consumption, an analysis of variance (ANOVA) test was used. The results show that the levels of 99 of 560 metabolites (17.7%, permutation test $P < 0.001$), 115 of 155 lipids (74.2%, permutation test $P < 0.001$) and

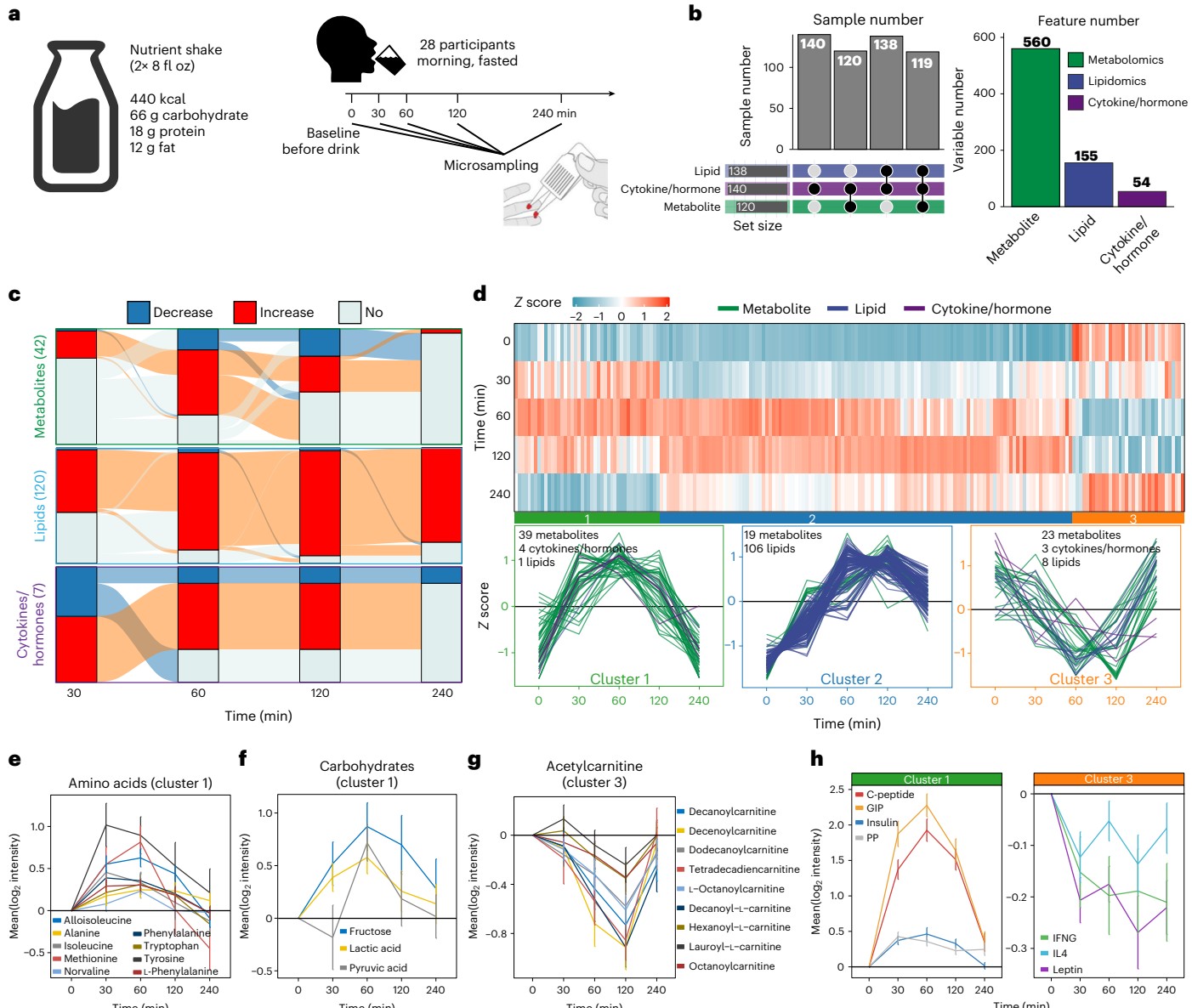

**Fig. 2 | The overview of Ensure shake study and molecular response to Ensure shake. a**, The study design and overview of the Ensure shake study. **b**, The summary of multi-omics data from the microsamples. **c**, Responses of metabolites, lipids and cytokines/hormones after Ensure shake consumption (two-sided Wilcoxon rank test). **d**, The clustering of dysregulated molecules following Ensure shake consumption. **e**, Amino acid response to Ensure shake consumption. **f**, Response of three dysregulated carbohydrates to Ensure shake consumption. **g**, Acylcarnitine response to Ensure shake consumption. **h**, Cytokine/hormone response to Ensure shake consumption. The points are represented by mean ± s.d.

7 of 54 cytokines/hormones (13.0%, permutation test $P < 0.001$) significantly shifted following Ensure shake consumption (Supplementary Dataset 4 and Methods). For the metabolites whose levels changed, the signals of analytes that differed from baseline were greater than those affected by storage duration. These results demonstrate that multi-omics analysis from microsamples can be used to measure the metabolic response to Ensure shake.

The molecules significantly affected by Ensure shake were then clustered using fuzzy $c$-means clustering to reveal and summarize the pattern of changes associated with consumption time (Methods). The shifted molecules were grouped into three major clusters across five timepoints (Fig. 2d). Cluster 1 contained 39 metabolites, 1 lipid and 4 cytokines that increased and then decreased with a peak at approximately 60 min following Ensure shake consumption and then returned to baseline by 240 min. Cluster 2 contained 19 metabolites and 106

lipids that increased more gradually than cluster 1, peaking at approximately 60–120 min. Molecules in cluster 3 decreased after consuming the Ensure shake and then recovered, including 23 metabolites, 8 lipids and 3 cytokines (Fig. 2d). These results demonstrate that the molecules have different patterns and kinetics of the biochemical responses to complex mixture ingestion.

## Altered metabolic pathway and physiological responses to Ensure shake

We next explored the pathways and physiological responses represented by the molecules in each cluster (Fig. 2d and Supplementary Fig. 2). Cluster 1 primarily comprised metabolites (39 metabolites, 1 lipid and 4 cytokines) and several biological pathways such as aminoacyl-tRNA biosynthesis, phenylalanine, tyrosine and tryptophan biosynthesis, and phenylalanine metabolism pathways were evident

(Supplementary Fig. 2c). The two major chemical classes captured in cluster 1 were amino acids and carbohydrates (Fig. 2d). Both compound classes probably come directly from the Ensure shake or are metabolized quickly (Fig. 2e,f). On the other hand, for cluster 3, acetylcarnitine was the main metabolite class, which dramatically decreased upon Ensure shake consumption and then recovered gradually by 240 min (Fig. 2g). This is expected because acetylcarnitine is broken down in the blood by plasma esterases to carnitine, and carnitine helps FFAs to be transported into the mitochondria for β-oxidation and energy production, hence maintaining whole-body energy homeostasis[17]. Consistent with this interpretation, eight FFAs detected in cluster 3 (Supplementary Fig. 2d) decreased following Ensure shake consumption. Notably, in cluster 2, we found 106 lipid species (Fig. 2d), and most of them were TAGs (102 TAGs with 48–52 carbons chains and 1–3 unsaturations; Supplementary Fig. 2e).

To better understand the molecules in the Ensure shake that might be directly detected in the participants' microsamples, we also analysed the composition of the Ensure shake using the same mass spectrometry procedure. Nearly 50% of the compounds found in the Ensure shake can be detected in the blood, and most of the remainder were of low abundance (Supplementary Fig. 2f). Importantly, of 21 high-interest metabolites that changed in the blood (Fig. 2e,f,g), 17 are present in the Ensure shake. This result demonstrates that the microsampling approach is able to detect the ingested molecular signatures from blood samples.

It is well known that both connecting peptide (C-peptide) and insulin are co-secreted from the pancreas and correlate with increased carbohydrates[18–20]. As expected, C-peptide and insulin were in the same cluster with the carbohydrates (cluster 1, Fig. 2h). Moreover, we found both gastric inhibitory polypeptide (GIP) and pancreatic polypeptide (PP) in the same cluster with insulin (cluster 1) (Fig. 2h, left). GIP is an inhibiting hormone of the secretin family of hormones[21], and its main role is stimulating insulin secretion[22]. Increased secretion of PP is reported to be associated with protein meal consumption, fasting, exercise and acute hypoglycemia[23]. In cluster 3, we found that leptin, interferon-γ (IFNG) and interleukin 4 (IL4) decreased quickly following Ensure shake consumption (Fig. 2h, right). The primary function of leptin is regulating adipose tissue mass through central hypothalamus-mediated effects on hunger[24]; its levels are expected to decrease after food consumption. IFNG and IL4 are involved in immune responses, including allergies and antibacterial responses. Interestingly, this suggests that the Ensure shake may have anti-inflammatory properties. In summary, these results demonstrate that the kinetics of the biochemical responses, including hormones, to complex mixture ingestion can be revealed using microsampling (Supplementary Dataset 5).

### Metabolic phenotyping reveals unique individual responses

How individuals respond to different foods is an area of great interest. The Ensure shake is a simple yet complex mixture of many types of simple molecules that can be quickly absorbed by the small intestine. To examine how different people respond to different metabolites, we explored the diversity in the kinetics and magnitude of the molecular responses among the different participants. Analysis of the samples using a t-distributed stochastic neighbour embedding (tSNE) plot shows that the samples were clustered by the participant, indicating that each participant had a unique molecular profile and that the difference between participants was greater than that of the effect of the shake (Extended Data Fig. 2a). Nonetheless, a clear timewise separation of data points was observed (Extended Data Fig. 2b). Our study suggested that, by 240 min, the metabolic levels tend to return closer to their baseline level (Extended Data Fig. 2b). We then used unsupervised consensus clustering to cluster participants into different groups. Our results suggested that there were two major groups based on the molecules altered in response to the shake consumption (Extended

Data Fig. 2c and Methods). In those two groups, we calculated the level of changes in metabolic features, comparing each timepoint with the baseline (timepoint 0) for each participant (Methods). This result also suggested that participants of those two groups had different responses to the Ensure shake (Extended Data Fig. 2d): group 2 responded more slowly than the participants in group 1 (Extended Data Fig. 2d), indicating the kinetics of their responses were different. Interestingly, for the 13 individuals with a measure of insulin resistance (steady-state plasma glucose (SSPG; Methods)), although statistically insignificant, we noticed a trend for patients with insulin resistance to be included in group 1 over group 2 (Wilcoxon test: $P = 0.29$, Extended Data Fig. 2e).

### Metabolic scores based on the dynamic response to the Ensure shake

As individuals are known to vary in their response to different foods, and we found heterogeneity in response to the Ensure shake for each participant, we next examined the response of each class of molecules, carbohydrates, lipids, cytokines/hormones and proteins to shake ingestion.

We derived a 'metabolic score' for the degree of an individual's carbohydrate, lipid, FFA and protein response to the Ensure shake, along with insulin secretion and inflammatory response (cytokines) (Methods). Briefly, for each molecule in each participant, after the Ensure shake consumption, the area under the curve (AUC) was used to represent its cumulative value (Fig. 3a). The AUCs of molecules for each molecular class (lipids, carbohydrates, amino acids and inflammatory molecules) were then used to calculate the response score for each participant (Fig. 3b). The final metabolic scores were normalized and ranged from 0 to 1, where 0 means the lowest relative metabolic level and 1 means the highest relative metabolic level. One participant was recognized as an outlier subject and excluded during the score calculation (Supplementary Fig. 3 and Methods). For each participant, we observed a consistent distribution pattern of the molecular species within each metabolic score indicative of similar response patterns to Ensure shake consumption. However, those patterns differed greatly across subjects demonstrating high inter-individual variability in the metabolism of nutrients (Supplementary Figs. 4 and 5a).

The six metabolic scores were calculated for each participant. As expected, we found a negative correlation between FFA score and SSPG, a marker of insulin resistance[25] (Supplementary Fig. 5b). Previous studies have demonstrated that elevated plasma levels of FFA are associated with insulin resistance[26]. The participants were classified into five groups on the basis of their metabolic scores using the hierarchical clustering method (Fig. 3c). We found that individuals varied considerably in their response to the shake for each of the different areas; examples selected from each of the five groups are shown in Fig. 3d. Within each group, we observed variations in the scores from the average score per metabolic class. For example, participant S30 in group 1 presented lower metabolic scores for fats and amino acids compared with the average level of the entire group. In comparison, S34 in group 5 showed higher scores for those classes (that is, carbohydrates and amino acids) than the average scores. These differences may be due to a variety of underlying mechanisms, including levels of digestive enzymes, transporters, hormones (such as incretins) and/ or intestinal microbes required to process particular molecules in the Ensure shake. Such underlying causes can be investigated in the future through additional analyses (such as metabolic flux analysis). Interestingly, S29 and S35 in groups 3 and 5, respectively, had higher scores in hormones and cytokines. The latter is particularly interesting as some individuals appear to have a strong inflammatory response (for example, individual S35), whereas others have a different response to appetite-suppressing hormones. Thus, the multi-omics data from microsamples reveal the enormous heterogeneity in the biochemical responses of each individual to a complex mixture. Such information can be defined using microsampling and is important for precision

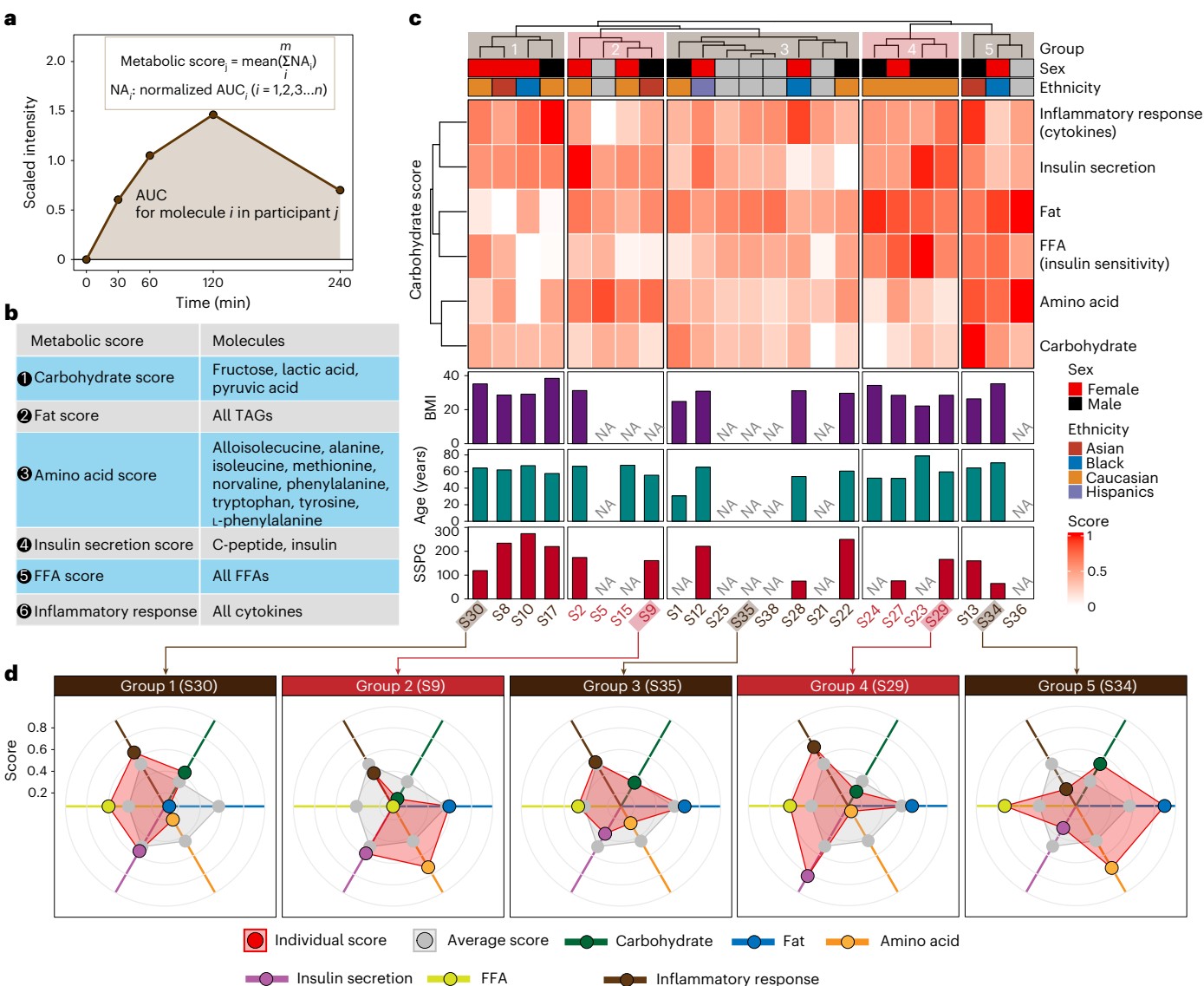

**Fig. 3 | Metabolic phenotyping based on the multi-omics response to the Ensure shake. a**, The visualization of the AUC metric for each analyte used in a metabolic score. **b**, The analytes used in calculating each of the six metabolic scores are shown. **c**, Participants were grouped into five groups on the basis of six metabolic scores. **d**, Five participant examples for each group.

nutrition diets, including inflammatory responses to food. Correlating these individual responses with medical phenotypes (for example, low-density lipoprotein levels and HbA1c levels) will be important for personalized nutrition management in the future.

**Case study 2: 24/7 personalized whole physiome profiling using wearable and multi-omics data**

Several studies have demonstrated that longitudinal individualized molecular profiles, clinical tests and digital data can monitor health and enable early disease detection at an individual level[1,27–29]. However, these studies use low-frequency/high-volume blood sampling (weekly or monthly, for instance), which does not enable the detection of detailed patterns such as circadian and many high-resolution lifestyle metabolic and other molecular changes. Higher-frequency data collection would enable monitoring of health status as well as circadian and lifestyle patterns at high resolution in real time, uncover relationships between molecules with each other and physiological and lifestyle activities and decipher causal associations between them at the personal level.

As a proof-of-principle study to determine whether this is feasible, we explored the combined use of our microsampling approach and wearables to explore the detailed molecular and physiological changes that occur in a real-world native context in a single individual. In this '24/7 study', a single participant collected blood microsamples usually every 1–2 h during waking hours over 7 days, with some samplings as short as 30 min apart (Fig. 4a and Supplementary Fig. 6a). A total of 98 samples were collected over 7 days along with wearable data from two devices: (1) a smartwatch that recorded heart rate (HR) and step count, and (2) a continuous glucose monitor (CGM)[30] (Fig. 4a, Supplementary Fig. 6b and Supplementary Dataset 6). Food logging was also performed many times each day using an app.

The 98 microsamples were used for in-depth multi-omics profiling, including untargeted proteomics, untargeted metabolomics, targeted lipidomics and targeted cytokine, hormone, total protein and cortisol assays (Fig. 4b, top). After data acquisition and annotation, we detected a total of 2,213 analytes that included 1,051 metabolites, 811 lipids, 291 proteins, 45 cytokines, 13 metabolic panels (cytokines/hormones), 1 total protein and 1 cortisol measurement (Fig. 4b, bottom)

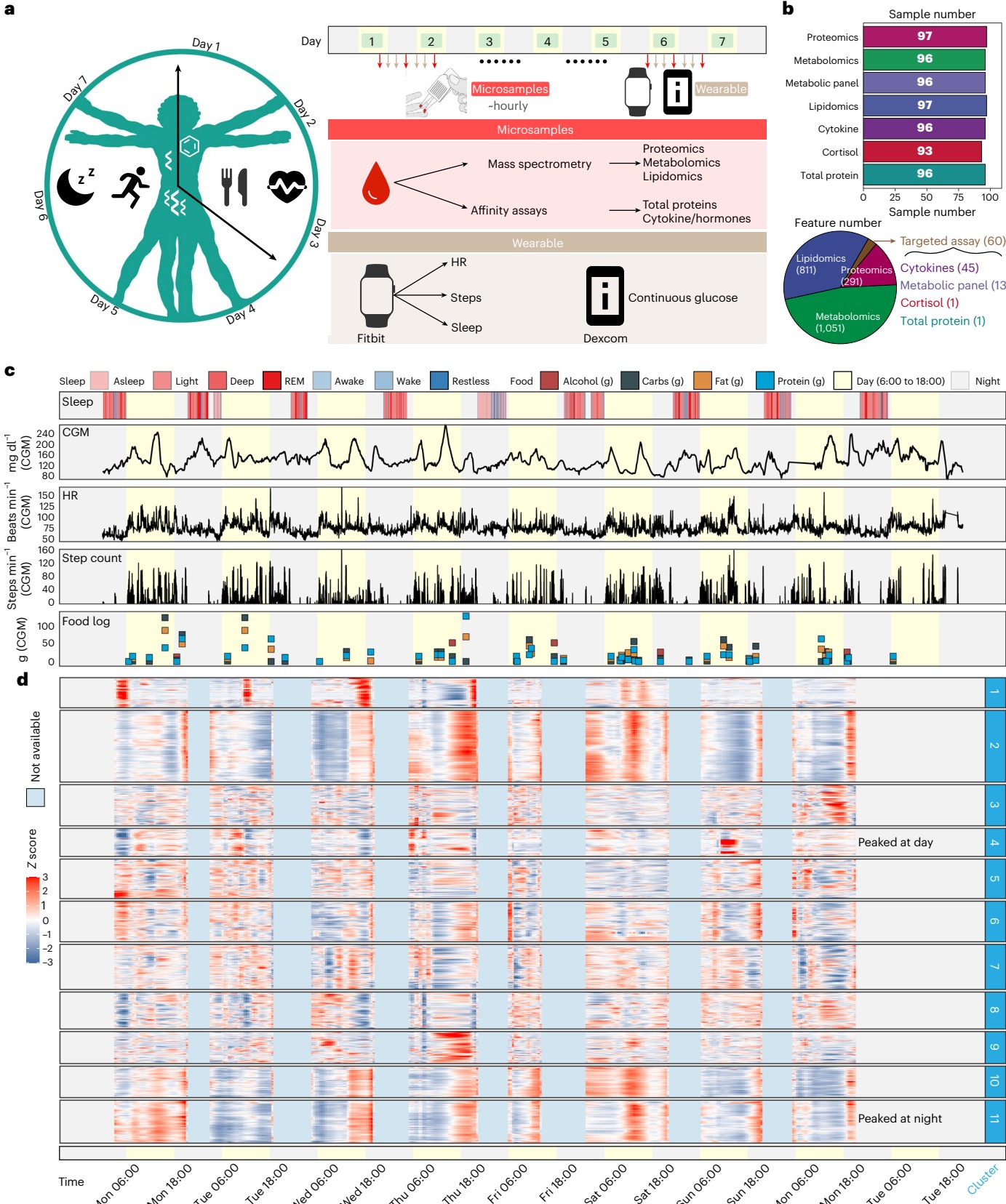

**Fig. 4 | Overview of the study design, sample collection and data acquisition for the 24/7 study. a**, One participant was closely monitored using wearable devices and high-frequency microsampling (approximately hourly) across 7 days. Microsamples were then analysed for internal multi-omics data measurements. **b**, Molecular information was detected from the high-frequency microsamples. **c**, Wearable data from the smartwatch (sleep and step count)

and Dexcom (CGM glucose). Legend defines the status of sleep (note that REM = rapid eye movement), the category of consumed foods, and the day/night period at every record. The yellow background represents the daytime (6:00 to 18:00). **d**, The internal molecules were grouped into 11 clusters using fuzzy *c*-means clustering.

resulting in a total of 214,661 biochemical measurements in addition to wearable physiological data (Supplementary Dataset 7). Overall, the prospective collection of internal molecular and wearable data resulted in comprehensive, high-frequency and abundant longitudinal data on the human whole physiome and lifestyle (Fig. 4b,c and Supplementary Fig. 6b,c), allowing us to explore how the internal molecules and physiology change on an hourly scale and their relationships at a personal level.

To explore whether multi-omics microsampling captures real biological signatures (such as food intake), we selected 2 days on which the participants ate high-carbohydrate food (131.8 g) and low-carbohydrate food (31.9 g), respectively (Supplementary Fig. 7a). Then the two carbohydrate metabolites (fructose and pyruvic acid) in microsamples were extracted and analysed, as shown in the box plot in Supplementary Fig. 7b. The median values of carbohydrate metabolites are 7.8 and 4.7, respectively, demonstrating that the omics data from microsamples roughly reflect the concentration of the food type the participants consumed.

### Wearable and internal multi-omics data reflect the individual physiological status

We first explored whether wearable and high-frequency internal multi-omics data can monitor and reflect the participant's health status and searched for general patterns in the data. The 2,213 internal molecular profiles were smoothed (Methods and Supplementary Fig. 7c,d) and then grouped into 11 clusters using fuzzy $c$-means clustering analysis (Fig. 4d and Extended Data Fig. 3a). Two clusters followed circadian patterns. For example, cluster 4, which is enriched by a high number of metabolites (Extended Data Fig. 3b), generally peaked during the day time, while cluster 11, which includes mostly lipids (Extended Data Fig. 3b), peaked primarily at night (Fig. 4d). Other clusters were not necessarily tied to circadian patterns and thus may reflect other events. The components of the different clusters were unique, indicating that the molecules have different temporal patterns (Extended Data Fig. 3b). To obtain tight and distinct molecular modules from each cluster, we used the community analysis method[31] (Methods and Extended Data Fig. 3c–e). Interestingly, obvious peaks were found in some modules (Extended Data Fig. 3e,f and Methods), indicating that the molecules in modules may be triggered by specific events (Figs. 4d and 5a).

As we have the detailed food (nutrition) and exercise logs, we next analysed whether and how molecular fluctuations relate to daily nutrition intake[32,33] (Methods). Briefly, nutrients in the food log were classified into several major classes on the basis of their content level: amino acids, vitamins, fat, electrolytes, calories, carbs and fibre. Next, we calculated the association between those classes with internal molecules presented by the Jaccard index depicted in the heat map (Extended Data Fig. 3g). Interestingly, we captured a high association between classes of amino acids and fat with several modules highly enriched in amino acids, FFAs and lipids (Extended Data Fig. 3g), consistent with previous results[34]. As with the Ensure shake study, our data revealed molecular associations with daily nutrition intake. For example, the participant consumed the same meal shake every morning during the study, and we captured a clear link between daily shake consumption and temporal increase of several compounds such as 1,2,3-benzenetriol sulfate and hydroxyphenyllactic acid, which are listed as the shake's ingredients (Fig. 5a, top, and Extended Data Fig. 3h).

Cortisol is believed to follow a circadian pattern, with levels higher in the morning that decrease towards the evening[35]. However, events during the day related to stress, activity and diet can impact cortisol levels[36]. Although morning peak levels of cortisol were evident on 3 days, we observed large day-to-day variations in cortisol patterns, demonstrating that within-day cortisol levels may not represent accurate inter-day cortisol patterns for this individual (Fig. 5a). This result suggests the importance of high-frequency sampling for monitoring health marker status.

Importantly, this study also demonstrates the potential usage of microsamples to measure the pharmacokinetics of a drug at an individual level. Our participant took a low dose of aspirin in the morning for 4 days. Microsampling accurately captured the pharmacokinetics of salicylic acid (hydrolysed product of aspirin, Extended Data Fig. 3h) and revealed a clearance period of about 24 h in this person, which is similar to previous results[37] (Fig. 5a, bottom). In addition, we found a negative correlation between caffeine and sleep quality (Extended Data Fig. 4a,b). This might be expected and has been reported in other studies[38,39]; however, the participant always consumed coffee before noon, indicating its long-lasting effect.

Interestingly, our detailed monitoring also revealed an unidentified inflammatory event in the middle of the week, spanning 3 days, with a number of both increased inflammatory cytokines (for example, TNFα and CD40L) and as well as several others that decreased (for example, eotaxin) (Extended Data Fig. 4c,d). This event was subclinical, as no symptoms were reported, and may represent an asymptomatic infection or other stress event. Together, these results show the power of high-frequency monitoring to record daily measures and health-related events not evident to the patient. The latter is particularly important for the early detection of disease[40].

### Circadian rhythms of internal molecules in human blood

Circadian rhythms are endogenous oscillators in physiological and behavioural processes over a 24 h cycle, and they play a critical role in human health and diseases[41]. Circadian molecules participate in diverse physiological phenomena such as cell division, energy metabolism and blood pressure[36,42]. These have not been explored at a personal level in a real-life setting because of the low frequency and high blood volume limitations of traditional blood sampling. Using the high-frequency data collected from the microsampling method, we were able to explore and evaluate molecules associated with circadian rhythms in the human body[43].

Each molecule was first searched for those that exhibited a consistent pattern across all 7 days, and we removed those that lacked a consistent daily pattern (Methods and Extended Data Fig. 5a). The circadian rhythms analysis (JTK_CYCLE algorithm[44]) was then used for quantitative analysis of all the molecules (Methods). We identified 332 circadian molecules (Benjamini–Hochberg (BH)-adjusted $P$ values < 0.05) that show clear circadian patterns (Extended Data Fig. 5b and Supplementary Dataset 8). The circadian molecules were grouped into five major clusters using fuzzy $c$-means clustering (Extended Data Fig. 5c). Interestingly, all clusters, except cluster 4 (enriched by protein), were dominated by lipids (Extended Data Fig. 5d). We focused on the molecules that exhibited a complete 1-day cycle (those in clusters 1, 2 and 3; Fig. 5b,c) and removed clusters 4 and 5, whose molecules had different levels at the beginning and end of the day (Extended Data Fig. 5e,f). Cluster 1 was dominated by PC (32.56%) and lysophosphatidylcholine (LPC, 25.58%), cluster 2 was dominated by TAGs (93.65%), and cluster 3 was dominated by both TAGs (49.15%) and phosphatidylethanolamine (PE, 22.03%) (Fig. 5d). Examples for each cluster are shown in Fig. 5e.

To explore the in-depth functions of the rhythmic molecules in each cluster, we performed lipid enrichment analysis using Lipid Mini-on[45]. LPC, PC, sterol and cholesterol ester (CE) were significantly enriched in cluster 1. Previous work has shown that LPC and PC have circadian rhythms with peak concentrations in the evening, consistent with our result[46]. For cluster 2, TAG and glycerolipid were significantly enriched, and for cluster 3, PE was significantly enriched (Extended Data Fig. 6). Thus, the different classes of lipids exhibit distinct circadian patterns. To explore whether the circadian lipids were affected by the food intake, we then examined the food logging data. We found that the fat nutrition intake differed across 8 days, meaning that the circadian lipids are not driven by the food intake. It is plausible that circadian lipids were driven by individual rhymic kinetics or gut microbes. In summary, multi-omics analyses from the high-frequency microsamples

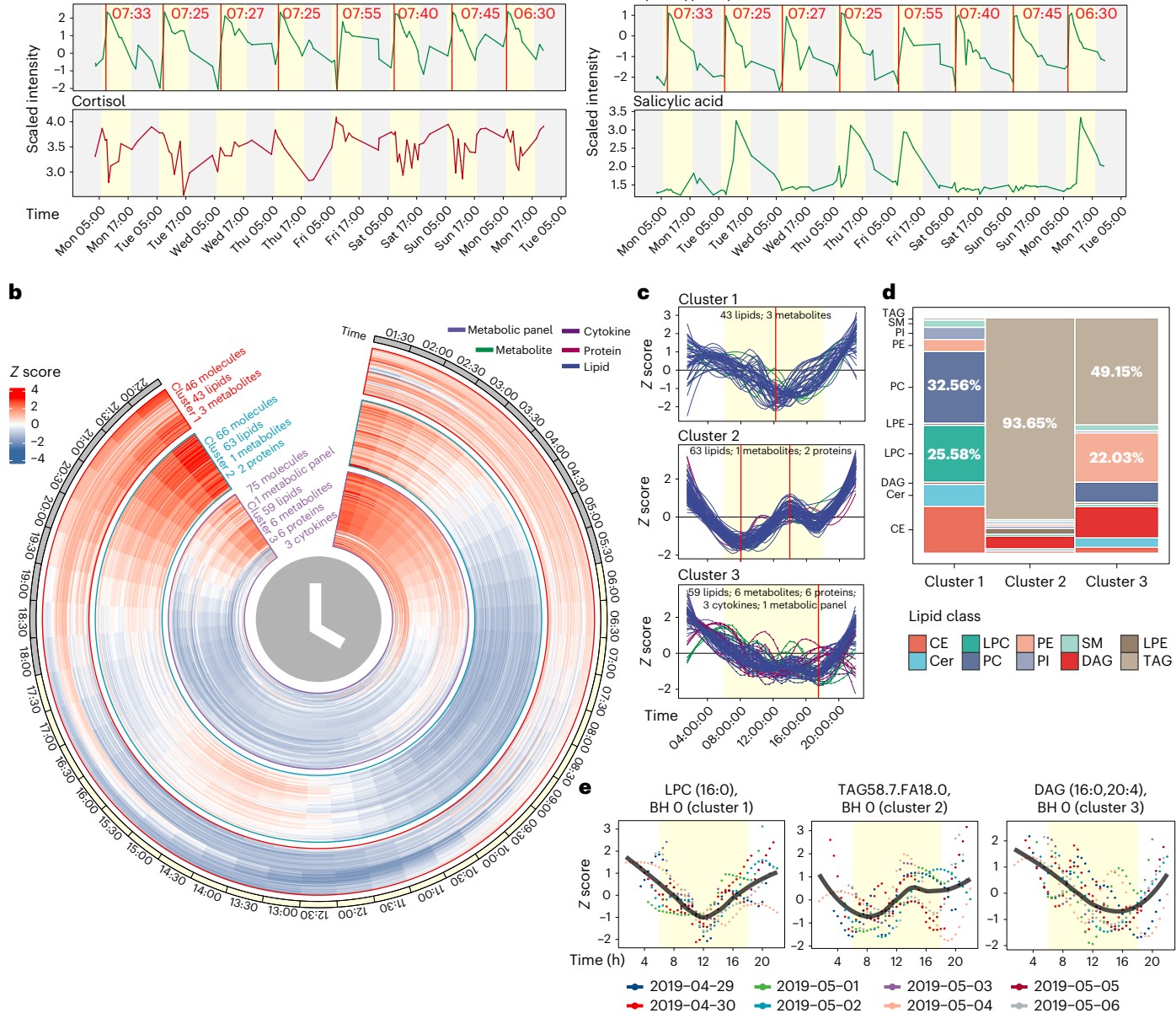

**Fig. 5 | Wearable and internal multi-omics data reflect the individual physiological status and circadian rhythm analysis of multi-omics data.** **a**, Four molecules reflect the participant's lifestyle. **b**, Heatmap to show the rhythmic molecules. **c**, Three clusters that have strong rhythmic patterns. **d**, Lipid class distributions of lipids in three clusters. Cer: ceramide; SM: sphingomyelin; DAG: diacylglycerol; LPE: lysophosphatidylethanolamine. **e**, Examples for each cluster. The yellow background represents the daytime (6:00 to 18:00).

revealed rhythmic molecules and demonstrated that lipids related to energy metabolism have distinct circadian patterns.

### Wearable data reflect internal molecular changes

Over the past several years, longitudinal monitoring of physiological data has garnered considerable interest[30,47–50]. However, the ability of wearable data to predict clinical labs has been limited[30]. Several studies have demonstrated that wearable data can reflect and predict the internal molecules (multi-omics data), including laboratory clinic tests and metabolites on a weekly or monthly scale[29,30]. However, due to the low-frequency sampling of multi-omics data, the circadian patterns and causal relationships between digital and internal molecular data cannot be discerned[50]. We explored the relationship between wearable data and internal molecular changes on an hourly scale at an individual level, including building predictive models.

Because of the different sampling frequencies of wearable and internal multi-omics data, we first attempted to match the wearable and internal multi-omics data using different window sizes. The matching windows were set as 5, 10, 20, 30, 40, 50, 60, 90 and 120 min. For each wearable data type (HR, step count and CGM) in the matched windows, a feature engineering pipeline[30] was used to convert different data types into eight features (for example, the standard deviation (s.d.) of heart standard and maximum HR; Methods) resulting in a total of 24 wearable features. The 24 wearable features were used to predict each analyte using the random forest model. Of the 2,223 molecules, we found 447 molecules that correlated with wearable features with at least one $R^2 > 0.3$ (Supplementary Fig. 8a). Interestingly, we also found that most molecules have higher prediction accuracy with the larger matching window, consistent with a previous study[30]. Most of the 447 molecules were lipids, and enrichment analysis showed that

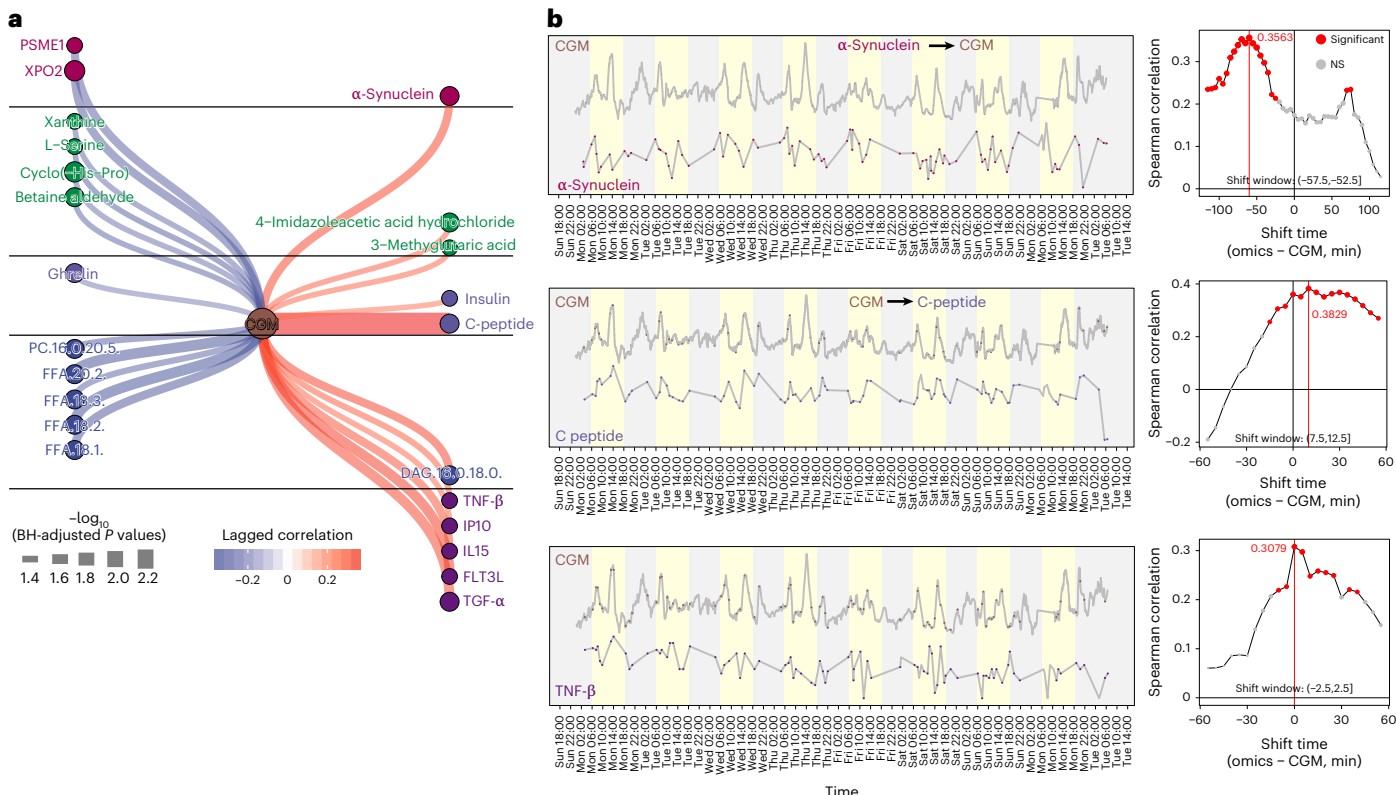

**Fig. 6 | CGM and internal molecule causal association network. a,** The CGM glucose subnetwork from the whole network. **b,** Three examples are shown to represent the causal relationships between CGM glucose and internal molecules. NS, not significant.

TAGs were the most predictable by wearable data (Supplementary Fig. 8b). HR-related features (for example, HR range, HR maximum and HR s.d.) contributed the most to the predictive models (Supplementary Fig. 8c). Using the random forest model, we then found that several cytokines (C-peptide, GIP, insulin and PP) could also be predicted by wearable data. The most contributed wearable features were CGM and HR-related features (Supplementary Fig. 8d). All those results demonstrate that wearable data could predict our high-frequency multi-omics data from microsamples.

Biochemical changes in the body can occur on the order of minutes and hours[51], and thus low-frequency multi-omics data (weekly or monthly) can find some associations with physiological measurements but not causal relationships[29,30]. Using the high-frequency microsampling approach, we next explored whether we could deduce the potential causal relationships between wearable data and internal molecules through temporal relationships; causal events are expected to precede downstream effects[52]. We first matched each wearable data point and molecule with different lagged times. Then, the Spearman correlation and P value were calculated for matching time-series data. Only the correlations with similar shapes and lagged time were scored as significant lagged correlations (Methods). To enable this analysis, the laggedCor (lagged correlation) algorithm was developed as an R package (https://jaspershen.github.io/laggedcor/). We then used this algorithm to demonstrate that we could capture and quantify the known lagged correlation and causal relationships between step count and HR. Interestingly, we found a lagged correlation of 0.6 (BH-adjusted P value < 0.0001) with a shift time of −1 min (step count − HR, Supplementary Fig. 9a), which means that 1 min after the step count increases, the HR begins to increase. This expected result demonstrates that our lagged correlation algorithm can capture and quantify potential causal relationships. Next, a lagged correlation network between wearable and internal molecular data was generated (Extended Data Fig. 7a

and Supplementary Fig. 9b), including 1,217 nodes (3 wearable data points and 1,214 molecules) and 1,895 edges (Extended Data Fig. 7b and Supplementary Dataset 9), demonstrating a high degree of association between wearable and multi-omics data. An example with the top 100 edges for each pair of wearable and omics data is provided in Extended Data Fig. 7a. Step count and HR have most of the edges (57.3% and 42.6%, respectively) in the lagged correlation network (Extended Data Fig. 7b). We also found that CGM correlates more with cytokines than HR and step count (Supplementary Fig. 9c), indicating that glucose levels strongly correlate with immune responses. This result has been demonstrated by other studies[53]. In addition, we also observed that step count and HR have many (669) overlapping correlations (Supplementary Fig. 9d), as expected since they have a significant positive correlation.

Interestingly, the immunity-related pathways contained some proteins that negatively correlated with CGM, which was not expected (Supplementary Fig. 9e). This demonstrates the importance of following these responses at the individual level. As expected, we also found that glucagon signalling, oxidative phosphorylation pathways and FFAs positively correlate with CGM (Supplementary Fig. 9f,g). Glucagon breakdown can raise the concentration of glucose and fatty acids in the bloodstream, and oxidative phosphorylation can oxidize nutrients to release chemical energy. We found that the caffeine metabolism pathway positively correlates with HR (Supplementary Fig. 9h), consistent with previous studies[54]. We also found that the blood coagulation pathway positively correlates with HR (Supplementary 14i), and the neutrophil degranulation pathway negatively correlates with HR (Supplementary Fig. 9j). To the best of our knowledge, these associations provide new biological insights that should be validated in future studies. Overview, these results demonstrate that the wearable data can reflect the physiological status of the participant and reveal useful insights at a personal level.

We extracted the CGM glucose subnetwork from the entire lagged correlation network to further explore how glucose associates with internal molecules (Fig. 6a). We observed that CGM glucose has a significant lagged correlation with α-synuclein (lagged correlation: 0.36, BH-adjusted P value < 0.05) (Fig. 6b), and the shift time is −55 min (α-synuclein − CGM), indicating that α-synuclein may directly or indirectly upregulate glucose levels in the blood. This result has been demonstrated by previous studies[55,56]. Previous studies have shown that higher C-peptide levels correlate with increased CGM glucose values[57]. Our data found that CGM glucose significantly lagged correlations with C-peptide (Fig. 6b and Supplementary Fig. 10a) and insulin (Supplementary Fig. 10b). The shift time between CGM and C-peptide in this individual is 10 min (lagged correlation 0.36, BH-adjusted P values < 0.05), which means that CGM glucose precedes the concentration of C-peptide in blood by 10 min. We also observed that CGM significantly correlates with several cytokines, including TNF-β (Fig. 6b), FLT3L, IL15, IP10 and TNF-α (Supplementary Fig. 10c; time shift 0 min to 15 min), and four of them are pro-inflammatory cytokines. These results indicate that glucose can cause a rapid and specific pro-inflammatory response. In summary, our results show that, on the basis of the high-frequency multi-omics data from microsamples, we find potential causal associations between wearable and multi-omics data. The potential causal relationships we found using the laggedCor algorithm can be validated by future experiments.

## Discussion

Current healthcare practices are reactive and based on limited physiological and/or clinical information, often collected months or years apart. In this study, we built a multi-omics microsampling approach that enables the measurement of thousands of metabolites, lipids, cytokines and proteins in frequently collected 10 µl blood samples. We demonstrated that many of the molecules from the microsamples (VAMS) are stable and reliable. In addition, most of the molecules from the microsampling are consistent with the classic blood sampling approach (Spearman correlation 0.8–0.9). Compared with DBS, VAMS can achieve good analytical performance for targeted compound and protein analysis[58,59]. However, for DBS, the haematocrit effect affects the resulting spot size, which can introduce variation in analysis. As the microsampling approach is less invasive and can be used remotely and without specific training, it enables high-frequency blood sample collection (approximately hourly) in a native setting, which is difficult to perform using the classic blood sampling approach. On the basis of the multi-omics microsampling workflow, we carried out two case studies to demonstrate the dense in situ samplings, and analytic capabilities to (1) perform dynamic and individualized metabolic assessments after response to dietary (Ensure shake) intervention and (2) reveal large-scale intra-day molecular fluctuations as well as thousands of molecular associations including those associated with intra-day variation in HR, glucose levels and activity.

It is worth noting that most analytes that we measure, particularly proteins, appear stable with regard to time, temperature and the combination of both. We also note that, since we tested for significant effects of storage conditions with a relatively low sample size, we do not rule out additional effects that may not have been observed here due to power challenges, which was evident from a sensitivity regression analysis that analysed only one storage condition at a time (storage duration and storage temperature), and additional effects for storage duration were identified when the baseline samples were added to the analysis. For those molecules that are not stable, they can either be discarded from the analyses or quantification can be ascertained from unique degradation products. Alternatively, sample collection procedures could include rapid and cold shipping to minimize potential issues with less stable molecules. Indeed, we found that most samples can be collected and stored within 24 h, thus minimizing degradation. Larger stability studies, especially in larger and more diverse populations, will help identify other potential issues. Regardless, reliable

measurements can be made for thousands of molecules, including those present at very low abundance (such as cytokines).

In summary, the presented methodology achieves fully remote, scalable, high-temporal-resolution omics and sensor monitoring. It has the potential for large-scale comprehensive, dynamic molecular and digital biomarker discovery and monitoring as well as health profiling. Here we used two case studies to show the potential of multi-omics microsampling in precision medicine. Many other applications can be envisioned. Examples include: (1) Longitudinal biomarker discovery. The multi-omics microsampling is simple and unpainful compared with the traditional blood collection method and thus enables anyone to self-collect high-frequent and high-quality blood microsamples anywhere for longitudinal biomarker discovery. (2) Personalized health monitoring. People can collect blood samples at home without any help and then send the samples to the laboratory for data acquisition and analysis. If a notable abnormality is detected, the result is sent immediately to a physician. The physician would then be able to validate the results and respond quickly with an intervention. (3) Therapeutic drug monitoring. Patients could collect microsamples frequently and remotely to monitor the drug-related compounds or biomarkers in the blood at a known time, to guide dosage, and result in optimized therapy. In our study, all the microsamples were prepared and run together as a batch to avoid batch effects. In the future, the microsamples collected in 1 day could be prepared and run in 1 day after sample collection. The users can receive their results within 2 days after sending their samples to the laboratory for analysis. Additionally, developing a clinical diagnostic based on microsampling requires additional validation steps for accuracy, precision, matrix effects and so on, and the use of standards such as isotopically labelled reference molecules. In addition, presently, only proteins, metabolites, lipids and cytokines were measured using our microsampling approach, but other types of molecules can be measured, such as DNA, epigenomes and RNA. For the 24/7 study, as a pilot study, only one participant was recruited to demonstrate the power of following personalized responses. Enlargement of the cohort size will enable the measurement of more generalized patterns but will also reveal new challenges in the processing and analysis of large numbers of samples. Indeed, our simple studies generated 98 data points in a single individual.

The two pilot case studies (group study and individual study) were used to demonstrate the power and application of the approach. The molecular signatures found in our study provide vast testable hypotheses that should be validated using analytical and experimental approaches. We note that group analysis is usually performed to find the overall trend. However, it can be potentially used to identify individual outliers who may have underlying conditions[1]. When an individual profile differs greatly from the average, one needs to first check for sample mix-ups, systematic variation and batch effects. Once normalized, data outlier detection can be further performed. Individuals who fall outside the overall pattern can be investigated for underlying causes for their molecular shift (medical conditions, medications or lifestyle abnormalities). In addition, the confounders (such as sex, age and body mass index (BMI)) must be controlled and adjusted to find the real and expected biological variation. Similarly, we note that, when an individual profile differs greatly from the average, overview conclusions from the whole cohort may not extend to individuals[33,60]. For the personalized analysis, the conclusion from the individual may not extend to the group or other individuals[60], which can be revealed using our approach. Overall, we believe the multi-omics microsampling approach offers a promising opportunity to integrate with wearable data to improve precision healthcare.

## Methods

### Microsampling blood sample collection
The Mitra device (Neoteryx) is used to collect the microsampling blood samples. The blood microsampling method and multi-omics data

acquisition workflow were established first (Fig. 1a). We developed a method for extracting proteins, lipids and metabolites from single microsamples, using biphasic extraction using MTBE. This extraction yields an organic phase processed for lipids, an aqueous phase processed for metabolites and a protein pellet processed for proteomics. Using a separate microsample, we performed an aqueous extraction for performing multiplexed immunoassays on the Luminex platform (Fig. 1a).

### Intravenous blood sample collection

Intravenous blood from the upper forearm was drawn from overnight-fasted participants. Specimens were immediately placed on ice after collection to avoid sample deterioration. Blood was collected in a purple top tube vacutainer (BD), layered onto Ficoll media (Thermo Fisher Scientific), and spun at 2,000 r.p.m. for 25 min at 24 °C. The top-layer EDTA–plasma was pipetted off, aliquoted, and immediately frozen at -80 °C. The peripheral blood mononuclear cell (PBMC) layer was collected and counted via the cell counter, and aliquots of PBMCs were further pelleted and flash frozen.

### Microsampling blood sample preparation

Mitra tip samples were thawed on ice, prepared and analysed randomly. Briefly, 300 µl of methanol spiked in with internal standards (provided with the Lipidyzer platform) was added to a Mitra tip and vortexed for 20 s. Lipids were solubilized by adding 1,000 µl of MTBE and incubated under agitation for 30 min at 4 °C. Phase separation was induced by the addition of 250 µl of ice-cold water. Samples were vortexed for 1 min and centrifuged at 14,000g for 5 min at 20 °C. The upper phase containing the lipids was then collected, dried down under nitrogen, reconstituted with 200 µl of methanol, and stored at −20 °C. After biphasic extraction, the Mitra tips were resuspended in 0.1 M Tris pH 8.6 buffer, along with 10% N-octyl-glucoside and 50 mM Tris(2-carboxyethyl)phosphine, followed by shaking at 60 °C for 1 h (denaturation, solubilization and reduction). The protein mixture was subsequently alkylated with 200 mM indole-3-acetic acid and incubated at room temperature (24 °C) in the dark for 30 min. Proteins were digested with trypsin overnight at 37 °C and quenched the following day with 10% (v/v) formic acid the following day. Three-hundred microlitres of metabolite layer was transferred, and then supplemented with 1,200 µl ice-cold MeOH:acetone:ACN (1:1:1) and vortexed for 10 s. The sample was incubated overnight at −20 °C. The samples were vortexed for 10 s, then centrifuged at 20,000g for 10 min at 4 °C. Then the sample was transferred to a new 2.0 ml tube and dried down. Finally, the samples were stored at −20 °C until data acquisition.

### Intravenous blood sample preparation

The sample preparation of venous blood samples for omics data acquisition is documented by our previous studies[1,2,29].

### Data acquisition of untargeted proteomics

Approximately 8 µg of tryptic digest were separated on a NanoLC 425 System (Sciex). A flow of 5 µl min⁻¹ was used with trap-elute setting using a ChromXP C18 trap column 0.5 × 10 mm, 5 µm, 120 Å (catalogue number 5028898, Sciex). Tryptic peptides were eluted from a ChromXP C18 column 0.3 × 150 mm, 3 µm, 120 Å (catalogue number 5022436, Sciex) using a 43 min gradient from 4% to 32% B with 1 h total run. Mobile phase solvents consisted of 92.9% water, 2% acetonitrile, 5% dimethyl sulfoxide and 0.1% formic acid (A phase) and 92.9% acetonitrile, 2% water, 5% dimethyl sulfoxide and 0.1% formic acid (B phase). Mass spectrometry analysis was performed using Sequential Window Acquisition of all Theoretical (SWATH) acquisitions on a TripleTOF 6600 System equipped with a DuoSpray Source and 25 mm inner diameter electrode (Sciex). Variable Q1 window SWATH acquisition methods (100 windows) were built-in high-sensitivity tandem mass spectrometry mode with Analyst TF Software (v1.7).

### Data processing of untargeted proteomics

The spectra were analysed with OpenSWATH using an in-house spectral library made from plasma and PBMC samples. Peak groups were then statistically scored with the PyProphet tool (v2.0.1), and all runs were aligned using the TRIC strategy. A final data matrix was produced with 1% false discovery rate (FDR) at the peptide level and 5% FDR at the protein level. Several QC steps were then applied to the output from SWATH2STATS. The correlation of peptide intensities between samples was calculated, and two samples with a mean sample correlation less than 2 s.d. from the mean sample correlation were removed. An additional sample with a peptide count less than 3 s.d. below the mean was removed. Poorly identified proteins and peptides were removed according to their $m$-scores using a target FDR of 0.05 ($m$-score threshold $8.91 \times 10^{-12}$). Peptides matched to an unknown protein, non-proteotypic peptides and peptides beyond the ten most intense peptides for a given protein were all removed. Protein intensities were then calculated by first summing the intensities of all transitions mapped to each peptide and then all peptides mapped to each protein. Proteins that were missing for > 50% of samples were removed, as were proteins whose CV among a separate set of three QC samples was greater than 50%. Each missing protein value was imputed using $k$-nearest neighbours (KNN; $k = 10$; using only non-imputed data; R package VIM, version 6.1.0). Protein values were then $\log_2$ transformed.

### Data acquisition of untargeted metabolomics

Prepared samples were analysed four times using hydrophilic interaction liquid chromatography (HILIC) and reverse phase liquid chromatography (RPLC) separation in both positive and negative ionization modes, respectively. Data were acquired on a Q Exactive Plus mass spectrometer for HILIC and a Q Exactive mass spectrometer for RPLC (Thermo Fisher Scientific). Both instruments were equipped with an HESI-II probe and operated in full mass spectrometry scan mode. Tandem mass spectrometry data were acquired on QC samples consisting of an equimolar mixture of all samples in the study. HILIC experiments were performed using a ZIC-HILIC column 2.1 × 100 mm, 3.5 µm, 200 Å (catalogue number 1504470001, Millipore) and mobile phase solvents consisting of 10 mM ammonium acetate in 50/50 acetonitrile/water (A phase) and 10 mM ammonium acetate in 95/5 acetonitrile/water (B phase). RPLC experiments were performed using a Zorbax SBaq column 2.1 × 50 mm, 1.7 µm, 100 Å (catalogue number 827700-914, Agilent Technologies) and mobile phase solvents consisting of 0.06% acetic acid in water (A phase) and 0.06% acetic acid in methanol (B phase).

### Data processing of untargeted metabolomics

Data from each mode were independently analysed using Progenesis QI software (v2.3, Nonlinear Dynamics). Metabolic features from blanks that did not show sufficient linearity upon dilution in QC samples ($r < 0.6$) were discarded. To reduce metabolic features of the metabolome profile, only metabolic features present in > 2/3 of the samples were kept for further analysis. Next, in the study samples, metabolic features present in > 50% of those samples were kept for further analysis. Missing values were imputed using KNN with $k = 10$. Data were then $\log_2$ transformed. The batch effect was evaluated using the dbnorm package[61]. Applying several batch removal algorithms, the ComBat model[62], giving the best performance, was considered for correcting systematic variation associated with the batch. Data from each mode were independently analysed using Progenesis QI software. ComBat was used to do data normalization[61], and KNN was used for missing value imputation. Data from each mode were merged, and metabolites were formally identified by matching fragmentation spectra and retention time to analytical-grade standards when possible or by matching experimental tandem mass spectrometry to fragmentation spectra in publicly available databases using metID[63]. We used the Metabolomics Standards Initiative[64] level of confidence to grade metabolite annotation confidence (levels 1 and 2).

### Data acquisition of semi-targeted lipidomics

Prepared samples were analysed using the Lipidyzer platform that comprises a 5500 QTRAP System equipped with a SelexION differential mobility spectrometry interface (Sciex) and a high-flow LC-30AD solvent delivery unit (Shimadzu). The detailed method can be found in our previous study[65]. In brief, lipid molecular species were identified and quantified using multiple reaction monitoring (MRM) and positive/negative ionization switching. Two acquisition methods were employed, covering ten lipid classes; method 1 had SelexION voltages turned on, while method 2 had SelexION voltages turned off. Lipidyzer data were reported by the Lipidomics Workflow Manager software, which calculates concentrations for each detected lipid as the average intensity of the analyte MRM/average intensity of the most structurally similar internal standard MRM multiplied by its concentration.

### Data processing of semi-targeted lipidomics

The final datasets were generated from the Lipidyzer platform, and the lipid abundances were reported as concentrations in nmol g$^{-1}$. Lipids detected in less than 2/3 of the samples were discarded, and missing values were imputed on the basis of a lipid class-wise KNN-TN (KNN truncation) imputation method[66].

**Cytokines and metabolic panel.** Cytokines were analysed using the HCYTMAG-60K-PX41 kit or the HSTCMAG28SPMX13 kit. For metabolic hormone assays, the catalogue number was HMHEMAG-34K. These assays were performed by the Human Immune Monitoring Center at Stanford University. All kits were purchased from EMD Millipore Corporation and used according to the manufacturer's instructions with the following modifications. Briefly, samples were mixed with antibody-linked magnetic beads on a 96-well plate and incubated overnight at 4 °C with shaking. Cold (4 °C) and room-temperature incubation steps were performed on an orbital shaker at 500–600 r.p.m. Plates were washed twice with wash buffer in a Biotek ELx405 washer. Following 1 h of incubation at room temperature with a biotinylated detection antibody, streptavidin–PE was added for 30 min with shaking. Plates were washed as described, and phosphate-buffered saline was added to wells for reading in the Luminex FlexMap3D Instrument (Thermo Fisher Scientific) with a lower bound of 50 beads per sample per cytokine. Each sample was measured in a singlet. Custom Assay Chex control beads were purchased from Radix BioSolutions and added to all wells.

**Cortisol.** This assay was performed by the Human Immune Monitoring Center at Stanford University using the ProcartaPlex Simplex Kit (catalogue number EPX010-12190-901, Thermo Fisher Scientific) and used according to the manufacturer's instructions with modifications as described. Briefly: Beads were added to a 96-well plate and washed in a BioTek ELx405 washer. Samples were added to the plate containing the mixed antibody-linked beads, and 20 µl of the competitive conjugate was added and incubated overnight at 4 °C with shaking. Cold (4 °C) and room-temperature incubation steps were performed on an orbital shaker at 500–600 r.p.m. Following overnight incubation, the plate was washed as described, and PE was added for 30 min at room temperature. The plate was washed as above, and a reading buffer was added to the wells. Each sample was measured in a single well. Plates were read using a Luminex FM3D FlexMap instrument with a lower bound of 50 beads per sample per cytokine. Custom Assay Chex control beads (Radix BioSolutions) were added to all wells.

**Total protein.** Total protein was determined by bicinchoninic acid assay according to kit instructions (Thermo Fisher Scientific).

**Wearable data.** The smartwatch (Fitbit Ionic) was used to collect the sleep, HR and step count data. The Fitbit Intraday API through the My Personal Health Dashboard app[67] was used to retrieve sleep, HR and step count data for the experiment period. The Dexcom G5 device was used to collect the CGM data. CGM data were transferred directly from the G5 device[51]. Dietary intake was logged manually using a notebook to track approximate meal timing and composition.

### Study design of stability analysis

All the microsamples were stored at −80 °C before they were prepared and analysed. The stability analysis was designed to explore whether the molecules from the microsamples are stable in different storage conditions (temperature and duration time) before they are stored at −80 °C. Two individuals were enroled under the institutional review board (IRB)-approved protocol (IRB-23602 at Stanford University) with written consent. By venepuncture, two individuals were asked to provide 10 ml of whole blood (in an EDTA purple top tube). The whole blood of each participant was poured into separate plastic reservoirs. Then 10 µl Mitra devices were touched to the surface of the blood to fill the microsample sponge. Thirty-six microsamples were generated for each participant, and microsamples were stored in duplicate at three temperatures (4, 25 and 37 °C) for six durations at the given temperature (3, 6, 24, 72, 120 and 0 h (that is, put into cold storage immediately)) before being stored at −80 °C until analysis. Then all the microsamples were prepared and used to acquire proteomics, metabolomics and lipidomics data using the protocol described above. All the omics data were provided as Supplementary Dataset 1.

### The first metric of stability

After the data generation, annotation, cleaning, imputation and transformations, each of the omic datasets (proteins, metabolite features and lipids) were assessed for analyte stability in storage. A total of 128 proteins (n = 66 samples), 1,461 metabolites (no redundant metabolite removal, n = 71 samples) and 776 lipids (n = 72 samples) were available for the stability analysis. The first metric assessed was the CV (estimated using the formula for log-transformed data[12]), which was calculated separately across all of the samples for each of the two participants from whom samples were taken. The mean of the two CVs (one from each participant's samples) was used as the CV for that analyte. The distribution of CVs was plotted.

### The second metric of stability

The second stability metric was used to identify storage conditions' significant effects on the analyte level. Linear regression was performed for each analyte where the analyte level was regressed on storage duration, temperature, the duration × temperature interaction effect, and an indicator for one of the two participants (to remove the effect of the actual difference in analyte level between the participants). As the samples that had 0 storage duration were never stored at any temperature, those samples were excluded from the analysis so that the effect of storage temperature could still be estimated, leaving 54, 59 and 60 samples for the protein, metabolite and lipid analyses, respectively. The 'lm' function in R was used, and since the objective of the study was to identify analytes that were stable under storage, a simple significance threshold of P = 0.05 was used to be more conservative since smaller P-value thresholds would exclude subtler potential effects of storage. The total model $R^2$ and the partial $R^2$ for each regression term were calculated using the 'rsq' and 'rsq.partial' functions of the 'rsq' package (version 2.2). The LMG measure of variable importance[1] was also calculated using the 'calc.relimp' function of the 'relaimpo' package (version 2.2-6). The proportion of statistically significant effects of storage conditions on analyte level was evaluated against the expected number of significant results at the alpha level of 0.05 to gauge the extent of signal for significant storage effects on the analytes. For each omic dataset and storage condition term, the top most associated analytes (according to P value) were plotted over time and coloured by storage temperature to visually examine the identified effects. As a lack of power might have prevented the identification of some storage

effects, each regression analysis was repeated but using two separate models, one testing only storage duration and one testing only storage temperature. The benefit of this change was that the baseline samples could be included in the models testing the effect of storage duration.

**Comparison between microsamples and intravenous plasma.** To compare the microsampling and conventional intravenous plasma collection approaches, 34 participants were enroled under the IRB-approved protocol (IRB-55689 at Stanford University) with written consent. Then one microsampling blood sample and one intravenous plasma sample were collected for each participant. All the samples were immediately saved at the −80 °C for subsequent sample preparation. Then all the samples were prepared and used to acquire untargeted metabolomics and lipidomics data according to the above protocols. For the metabolomics data, after data processing and data curation, 22,858 metabolic features were detected (RPLC positive mode: 7,487 features, RPLC negative mode: 4,662 features, HILIC positive mode: 6,362 features, HILIC negative mode: 4,374 features). Only 642 features with annotations (Metabolomics Standards Initiative levels 1 and 2) remained for subsequent analysis. For the lipidomics data, 616 lipids were detected. All the omics data are provided in Supplementary Dataset 2.

**Ensure shake study cohort.** Twenty-eight participants were enroled in the Ensure shake study under the IRB-approved protocol (IRB-47966 at Stanford University) with written consent. Twenty-one out of 28 participants have completed demographic data (Supplementary Fig. 2). The median SSPG is 166, the median age is 64.2 years, and the median BMI is 29.7 kg m$^{-2}$. Among all the participants, 38% are male, 14.3% are Asian, 14.3% are Black, 66.7% are Caucasian and 4.8% are Hispanic. All 28 participants were mailed a kit containing microsampling devices (Mitra device), Ensure shake (contains 440 kcal, 66 g carbohydrate, 18 g protein and 12 g fat) and instructions for the microsampling sample collection. Each participant was instructed to consume the Ensure shake and then collected microsampling blood samples immediately before consuming Ensure shake (baseline, timepoint 0), and at 30, 60, 120 and 240 min following Ensure shake consumption (Supplementary Fig. 2b). Finally, we collected five timepoint microsamples for each participant (Supplementary Fig. 2b). Participants were asked to return their microsamples by overnight mail the same day after blood sample collection. Then all the microsamples were used for multi-omics data acquisition, namely, untargeted metabolomics, targeted lipidomics and cytokine/hormone. Four participants (S6, S26, S31 and S37) without metabolomics data were removed from the final dataset (Supplementary Fig. 2b). After data cleaning, curation and annotation, 768 analytes were detected from the microsamples, containing 560 metabolites, 155 lipids and 54 cytokines/hormones. All the omics data are provided in Supplementary Dataset 3.

**24/7 study cohort.** Only one participant (male, 64 years old) was enroled in the 24/7 study under IRB-approved protocol (IRB-23602 at Stanford University) with written consent. The microsampling method enables frequent sampling on the order of minutes or hours. However, to make it acceptable and executable, the participant was instructed to perform self-collected finger prick microsamples approximately every hour during waking and every two hours during overnight periods sporadically for 7 days (Fig. 4a and Supplementary Fig. 6a). In addition, the participant was also instructed to leverage several wearable devices (Fitbit smartwatch, Dexcom) to acquire comprehensive digital data (wearable data), including the HR, step count, CGM and food logging. The microsamples were immediately saved on dry ice upon collection by the participant and then shipped to the laboratory daily. Finally, 97 microsamples in total were collected. They were used to perform in-depth multi-omics data acquisition, including (1) untargeted proteomics, (2) untargeted metabolomics, (3) semi-targeted lipidomics and (4) targeted assay (cytokine, hormones, total protein

and cortisol). After data processing, curation and annotation, from the microsamples, we finally detected a total of 2,213 analytes that included 1,051 metabolites, 811 lipids, 291 proteins, 45 cytokines, 13 metabolic panels (cytokines/hormones), 1 total protein and 1 cortisol. All the data are provided as a resource in Supplementary Datasets 6 and 7.

**General statistical, bioinformatics analysis and data visualization.** Most statistical analysis and data visualization were performed using RStudio and R language (version 4.1.2). Most of the R packages and their dependencies used in this study are maintained in CRAN (https://cran.r-project.org/) or Bioconductor (https://bioconductor.org/). The detailed version of all the packages can be found in Supplementary Note. The main script for analysis and data visualization is provided on GitHub (https://github.com/jaspershen/microsampling_multiomics).

In general, before all the statistical analysis, the data are log$_2$ transformed and then auto-scaled. All the multiple comparisons were adjusted by the BH method using the 'p.adjust' function in R. The R functions 'cor' and 'cor.test' were used to calculate the Spearman correlation coefficients. The R package 'ggplot2' was used to perform most of the data visualization in this study. The R package 'Rtsne' was used for the tSNE analysis in the Ensure shake study. The icons used in figures are from iconfont.cn, which can be used for uncommercial purposes under the MIT license (https://pub.dev/packages/iconfont/license).

**Differentially expressed molecules after consuming Ensure shake.** In the Ensure shake study, the timepoint 0 (before consuming Ensure shake) was set as the baseline, and all the other four timepoints were compared with the baseline to get the differentially expressed molecules (metabolites, lipids and cytokines/hormones). The paired Wilcoxon rank-sum test ('wilcox.test' function of R) was used to get the P values. The multiple comparisons were adjusted using the BH method ('p.adjust' function of R). And the adjusted P values less than 0.05 were considered as significantly differentially expressed molecules. Then the number of significant molecules whose level had changed at different timepoints was visualized using a Sankey plot ('ggalluvial' package of R). Next, after consuming Ensure shake across all the timepoints, we identified the entire set of molecules whose levels changed. The ANOVA test ('anova_test' function from the 'rstatix' package in R) was used to calculate the P values and then adjusted using the BH method. To evaluate whether the significantly expressed molecules we found were random or not, a permutation test was performed. In brief, the sample labels of omics data were randomly shifted to get the random datasets. Then the same method (ANOVA test) was used to find the altered molecules for the random dataset. This step was repeated 100 times to get a null distribution of differential molecules. Then the permutation P value was calculated to evaluate whether the expressed molecules were random.

**Consensus clustering.** In the Ensure shake study, the unsupervised k-means consensus clustering of all samples was performed with the R packages 'CancerSubtypes' and 'ConsensusClusterPlus' using the significantly shifted molecules that were discovered after consuming the Ensure shake[68]. The data were log$_2$ transformed first and then auto-scaled. Samples clusters were detected on the basis of k-means clustering, Euclidean distance and 1,000 resampling repetitions in the 'ExecuteCC' function in the range of two to six clusters. The generated empirical cumulative distribution function plot initially showed the optional separation of two clusters for all samples. To further decide how many groups (k) should be generated, the silhouette information from clustering was extracted using the 'silhouette_SimilarityMatrix function'. We compared k = 2, 3, 4 and 5 and found that, when k = 2, we got high stability for clustering (Extended Data Fig. 2c). From the consensus matrix heat maps, two groups seem to have the best clustering (Extended Data Fig. 2d). So finally, all the samples were assigned to two groups.

**Fuzzy *c*-means clustering.** The R package 'Mfuzz' was used for fuzzy *c*-means clustering[69]. In brief, the omics data were first $\log_2$ transformed and auto-scaled, and then the minimum centroid distances were calculated for cluster numbers from 2 to 22 by step 1. The minimum centroid distance is used as the cluster validity index. Then the optimal cluster number was selected according to rule[70]. To get a more accurate cluster number, the clusters whose centre expression data correlations are more than 0.8 were merged as one cluster. Then the optimal cluster number was used to do the fuzzy *c*-means clustering. For each cluster, only the molecules with memberships of more than 0.5 were retained for subsequent analysis.

**Metabolic scores.** Participant S18 was considered as an outlier in the baseline and removed from the dataset for subsequent analysis (Supplementary Fig. 3). Then five metabolic scores were calculated: (1) Three carbohydrates (fructose, lactic acid and pyruvic acid) were detected and used to calculate the carbohydrate score, which represents the human's ability to metabolize carbohydrates (Supplementary Fig. 4). (2) Nine amino acids (alloisoleucine, alanine, isoleucine, methionine, norvaline, phenylalanine, tryptophan, tyrosine and L-phenylalanine) were detected and used to calculate the amino acid score (protein), which represents the human's ability to metabolize proteins (Supplementary Fig. 4). (3) A total of 103 TAGs were detected and used to calculate the fat score, representing the human's ability to metabolize the fat (Supplementary Fig. 4). (4) The C-peptide and insulin were detected and used to calculate the insulin secretion score, representing the human's ability to secrete insulin (Supplementary Fig. 4). (5) The eight FFAs (FFA 16:0, FFA 16:1, FFA 18:1, FFA 18:2, FFA 18:3, FFA 22:2, FFA 22:5 and FFA 22:6) were detected and used to calculate FFA (insulin sensitivity) score, which represents the human's ability to respond to insulin sensitivity (Supplementary Fig. 4). (6) All the cytokines were used to calculate the immune response score representing the human's immune response (Supplementary Fig. 5a).

For each metabolic score MS, the molecules $M_i$ ($i = 1, 2, 3 \ldots m$) in this group were first defined and selected (Fig. 3b), and then the dataset was $\log_2$ transformed and auto-scaled. For each participant and molecule, the intensity values across all the timepoints were subtracted by the baseline value, so the baseline value was 0. Then the AUC $A_{i,j}$ was calculated for molecule $M_i$ ($i = 1, 2, 3 \ldots m$) and participant $P_j$ ($j = 1, 2, 3 \ldots n$). To normalize the $A_{i,j}$, the $A_{i,j}$ were subtracted by the minimum $\min(A_{i,j})$ and divided by the range of all the AUCs ($\max(A_{i,j}) - \min(A_{i,j})$). The normalized $A_{i,j}$ is labelled as $NA_{i,j}$ and is from 0 to 1. Then, each metabolic score $MS_j$ in each participant $j$ is calculated as below:

$$MS_j = \text{mean}\left(\sum_i^m NA_i\right)$$

where $MS_j$ is the metabolic score for participant $j$, and $NA_i$ is the normalized AUCs of molecule $i$ ($i = 1, 2, 3 \ldots m$). For the carbohydrate score, amino acid (protein) score, fat score and FFA score (insulin sensitivity), the high AUCs of molecules mean that the person's ability to metabolize the molecules is low, so the final metabolic scores were calculated as $1 - MS_j$. For the insulin secretion score and immune response score, the final score is the same as the $MS_j$.

**Metabolomics pathway enrichment**
To do the metabolomics pathway enrichment, the human KEGG pathway database was downloaded from KEGG using the R package massDatabase[71]. The original KEGG database has 275 metabolic pathways. Then we separated them into metabolic pathways or disease pathways on the basis of the 'class' information for each pathway. The pathways with the 'human disease' class were assigned to the disease pathway database, which contains 74 pathways, and the remaining 201 pathways were assigned to the metabolic pathway database. The pathway enrichment analysis is used in the hypergeometric distribution test from the

tidyMass project[72]. The BH method was used to adjust $P$ values, and the cut-off was set as 0.05 (BH-adjusted $P$ values < 0.05).

**Lipidomics data enrichment analysis**
The Lipid Mini-on software was used to do the lipid enrichment analysis[45]. In brief, the lipids' names were first modified to meet the requirement of the tool. The dysregulated lipids were uploaded as query files, and all the detected lipids were uploaded as universe files. The default Fisher's exact test was used as the enrichment test method. The category, main class, subclass, individual chains, individual chain length and number of double bonds were selected for general parameters to test. Finally, the enrichment result containing detailed tables and networks was downloaded for subsequent analysis.

**Proteomics pathway enrichment**
The R package 'clusterProfiler' was used for proteomics pathway enrichment. We first converted the gene ID of proteins to ENTREZID ID, and then the Gene Ontology (GO) database was used for GO term enrichment analysis. The $P$ values were adjusted using the BH method, and the cut-off was set as 0.05. Only the enriched GO terms with at least mapped five proteins remained to ensure that the enriched GO terms have enough genes. To reduce the redundancy of enriched GO terms, the similarity between GO terms was calculated using the 'Wang' algorithm from the R package 'simplifyEnrichment'[73]. And only the connections with similarities > 0.3 remained to construct the GO term similarity network. Then the community analysis (R package 'igraph') was used to divide this network into different modules. The GO term with the smaller enrichment adjusted $P$ values was selected for each module as the representative.

**LOESS smoothing data.** In the 24/7 study, the timepoints of microsamples for each day differ. However, the circadian analysis requires enough timepoints for each day. So we leveraged the locally estimated scatterplot smoothing (LOESS) method to smooth and predict the multi-omics data in the specific timepoints (every half hour) described in another publication[74]. In brief, for each molecule, we fitted it with the LOESS regression method for each day ('loess' function in R). During the fitting, LOESS's argument 'span' was optimized by cross-validation. As the gap between 2 days is always more than 4 h, we did not fit the time between 2 days for an accurate and robust fitting and prediction. After getting the LOESS prediction model, we predicted each molecule's intensity every half hour during the days.

**Correlation network and community analysis.** In the 24/7 study, we constructed a correlation network for each cluster that we got using fuzzy *c*-means clustering. In brief, the Spearman correlation was calculated for every two molecules. Only the correlations with coefficient > 0.7 and BH-adjusted $P$ values < 0.05 remained for subsequent analysis. All the remained correlations were used to construct the correlation network. To get more accurate and distinct modules, we use the community analysis to extract subnetworks (modules) from the correlation network[31]. Here we used the fast greedy modularity optimization algorithm ('cluster_fast_greedy' function from the R package 'igraph'). Finally, 11 clusters and 83 modules were detected. The R packages 'igraph' and 'ggraph' were used to visualize the network.

**Associations between molecular modules and nutrition intake.** In the 24/7 study, to evaluate the associations between molecular modules and nutrition intake, peak detection (Gaussian distribution fitting) was first used to find the 'peaks' in each module (Extended Data Fig. 3f). If there is a peak, then it is marked as '1' at this time. If not, it is marked as '0'. For food, if the participant consumes this food at this timepoint, then this timepoint will be marked as '1' for this food. Then, for each food and module, the Jaccard index was calculated, and

only the pairs with a Jaccard index > 0.3 were retained for subsequent analysis (Extended Data Fig. 3g).

**Consistency score for molecules.** In the 24/7 study, the consistency score was designed and calculated for each molecule to assess whether one molecule is consistent daily. LOESS smoothed data was used for consistency score calculation. For each molecule, the Spearman correlations between 2 days were calculated, and the median correlation value was calculated and considered as the consistency score for this molecule. Only the molecules with consistency scores > 0.6 were retained for the next circadian analysis.

**Circadian rhythm analysis.** In the 24/7 study, the R package 'MetaCycle' is used to do the circadian rhythm analysis[43]. The LOESS smoothed omics data were $\log_2$ transformed and auto-scaled. Then, the times for samples were set as the timepoints in the 'meta2d' function. The Lomb–Scargle was selected for circadian rhythm analysis[75]. The $P$ values were adjusted using the BH method. Only the molecules with BH-adjusted $P$ values < 0.05 were considered statistically significant circadian molecules and retained for subsequent analysis.

**Wearable data predicts internal molecules.** In the 24/7 study, to evaluate whether the wearable data could be used to predict internal molecules, the method from a previous publishment[30] was used. As the frequency of wearable data and internal molecules are different, we need to match the internal molecule and wearable data first. The matching windows were set as 5, 10, 20, 30, 40, 50, 60, 90 and 120 min, respectively. For the wearable data points that matched with internal molecules, a feature engineering pipeline[30] was used to convert the wearable data into eight features: mean value, median value, standard, maximum, minimum, skewness, kurtosis and range. So, each wearable data point was converted into eight features. The wearable data (HR, step count and CGM) were converted to 24 features in total and were used as independent variables to predict each internal molecule. The random forest model (R package 'caret' and 'RandomForest'), which has been proven to have the best prediction accuracy, was used[30]. The 24 wearable features were combined for each internal molecule to construct the prediction model. The sevenfold cross-validation method was used during the prediction model construction. The importance of each wearable feature was saved for subsequent analysis.

**Lagged correlation.** In the 24/7 study, to calculate the lagged correlation between wearable data and internal molecules, we have developed the laggedCor algorithm (lagged correlation) and an R package named 'laggedcor' (https://jaspershen.github.io/laggedcor/). The laggedCor algorithm can be used to extract potential causal relationships. Let us assume that $X$ is wearable data and $Y$ is internal omics data. In a real biological system, if $X$ and $Y$ have a causal relationship ($X$ causes $Y$), $Y$ often responds to $X$ after a certain lapse of time. Such a lapse of time is called a lag time. This means that $X$ and $Y$ change asynchronously. To explore whether $X$ and $Y$ have a potential causal relationship, we just shift the lag time between $X$ and $Y$ for matching and then calculate the correlation between them. Suppose the $X$ and $Y$ have a potential causal relationship and the lag time is $T$; then we can get the highest lagged correlation between $X$ and $Y$ at the lag time $T$.

Briefly, two time-series data are used as the inputs for *laggedcor*. The lower frequency time-series data (in the 24/7 study, the omics data) are labelled as $X_t$ ($t \in$ Ti), and the higher frequency time-series data (in the 24/7 study, the wearable data) are labelled as $Y_t$ ($t \in$ Tj). To make sure that there are overlaps between $X_{ti}$ and $Y_{tj}$, they should meet the below equation:

$$T_i \cap T_j \neq \varnothing$$

Then the two series data, $X_t$ and $Y_t$, are used to calculate the lagged correlation as described in the steps below.

**Step 1: matching between $X_t$ and $Y_t$.** Every sample point $Y_{tj}$ in $Y$ is used to match the sample points in $X_t$. The shift time is labelled as Ts (Ts is set on the basis of the frequency of $X_t$ and $Y_t$), and the matching time window is labelled as Tw. So the sample points $X_{ti}$ in $X_t$ that meet the below equation are labelled as matched sample points for $Y_{tj}$ in $Y$:

$$tj + \mathrm{Ts} - \frac{\mathrm{Tw}}{2} \leq ti < tj + \mathrm{Ts} + \frac{\mathrm{Tw}}{2}; i \in (1, 2, 3 \ldots m)$$

Then the matched sample points $X_{ti}$ are averaged as $X_{tj}$ that matched with $Y_{tj}$ in $Y$:

$$X_{tj} = \sum_{ti}^{tm} X_{ti}$$

Then we get the new time-series data $X_t$ ($t \in$ Tj).

**Step 2: correlation calculation.** Then the Spearman correlation between $X_t$ and $Y_t$ ($t \in$ Tj) is calculated with the shift time Ts. And the correlation rho and $P$ value are recorded as $\mathrm{Cor}_{ts}$ and $p_{ts}$.

**Step 3: repeat step 1 and step 2 with different shift time.** Then, step 1 and step 2 are repeated for a series shift times $\mathrm{Ts}_i$, $i = 1, 2, 3 \ldots n$; $\mathrm{Ts}_1 < 0$ and abs($\mathrm{Ts}_1$) = abs($\mathrm{Ts}_n$). Then we can get a series $\mathrm{Cor}_{ts}$ and a series $p_{ts}$, ts $\in$ Ts.

**Step 4: evaluation of the significance of lagged correlation.** The maximum correlation of $\mathrm{Cor}_{ts}$ and related $P$ value are extracted as the lagged correlation for time-series data $X_t$ and $Y_t$. To evaluate whether the lagged correlation is significant, the Gaussian distribution is used to fit the $\mathrm{Cor}_{ts}$, and the correlations in all the shift times are calculated using the fitted Gaussian distribution and labelled as $\mathrm{PCor}_{ts}$. The quality score was then calculated as the absolute Spearman correlation score between $\mathrm{PCor}_{ts}$ and $\mathrm{Cor}_{ts}$. Only the lagged correlation with a quality score was considered a real lagged correlation and used for subsequent analysis.

### Reporting summary

Further information on research design is available in the Nature Portfolio Reporting Summary linked to this article.

### Data availability

All the analysed data used in this study are provided as a supplementary dataset. Source data are provided with this paper.

### Code availability

R version 4.1.2 was used with the base packages and other packages, and detailed information is provided in Supplementary Information (section Supplementary Note). All the custom scripts for data analysis and data visualization are provided open-source via https://github.com/jaspershen/microsampling_multiomics and Zenodo (https://zenodo.org/record/7393012#.Y4sEj-yZP0o). The laggedCor algorithm and package were developed for lagged correlation calculation and are available open-source via https://jaspershen.github.io/laggedcor.

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

## Acknowledgements

M.P.S. discloses support for the research described in this study from the National Institute of Health (grant numbers 5RM1HG00773508 and 5R01AT01023204). Special thanks to Benjamin Rolnik for his various contributations on sample collection and funding application in this study.

## Author contributions

R.K., D.H., X.S. and M.P.S. conceived and designed the study; D.H., B.L.-M., K.E.C., B.M., I.S., S.A., A.G., K.C. and R.K. prepared samples and acquired lipidomics, metabolomics and proteomics data; Y.R.-H. prepared samples and generated Luminex data. D.J.P., N.B. and X.S. performed the stability analysis. X.S. and R.K. analysed the data of the Ensure shake study; X.S., R.K., J.U., A.D. and C.W. analysed the data of the 24/7 study. X.S. and C.W. developed the laggedCor algorithm and built the R package. X.S., C.W. and D.J.P. prepared all the figures. X.S., R.K., N.B., D.H., D.J.P., C.W. and M.P.S. wrote the manuscript. All the authors contributed to the final version of the manuscript.

## Competing interests

M.P.S. is a co-founder and scientific advisor of Personalis, SensOmics, Qbio, January AI, Fodsel, Filtricine, Protos, RTHM, Iollo, Marble Therapeutics and Mirvie. He is a scientific advisor of Genapsys, Jupiter, Neuvivo, Swaza and Mitrix. D.H. has a financial interest in Seer Inc. and Prognomiq Inc. R.K. is a co-founder of RTHM Inc. All other authors declare no competing interests.

## Additional information

**Extended data** is available for this paper at https://doi.org/10.1038/s41551-022-00999-8.

**Correspondence and requests for materials** should be addressed to Michael P. Snyder.

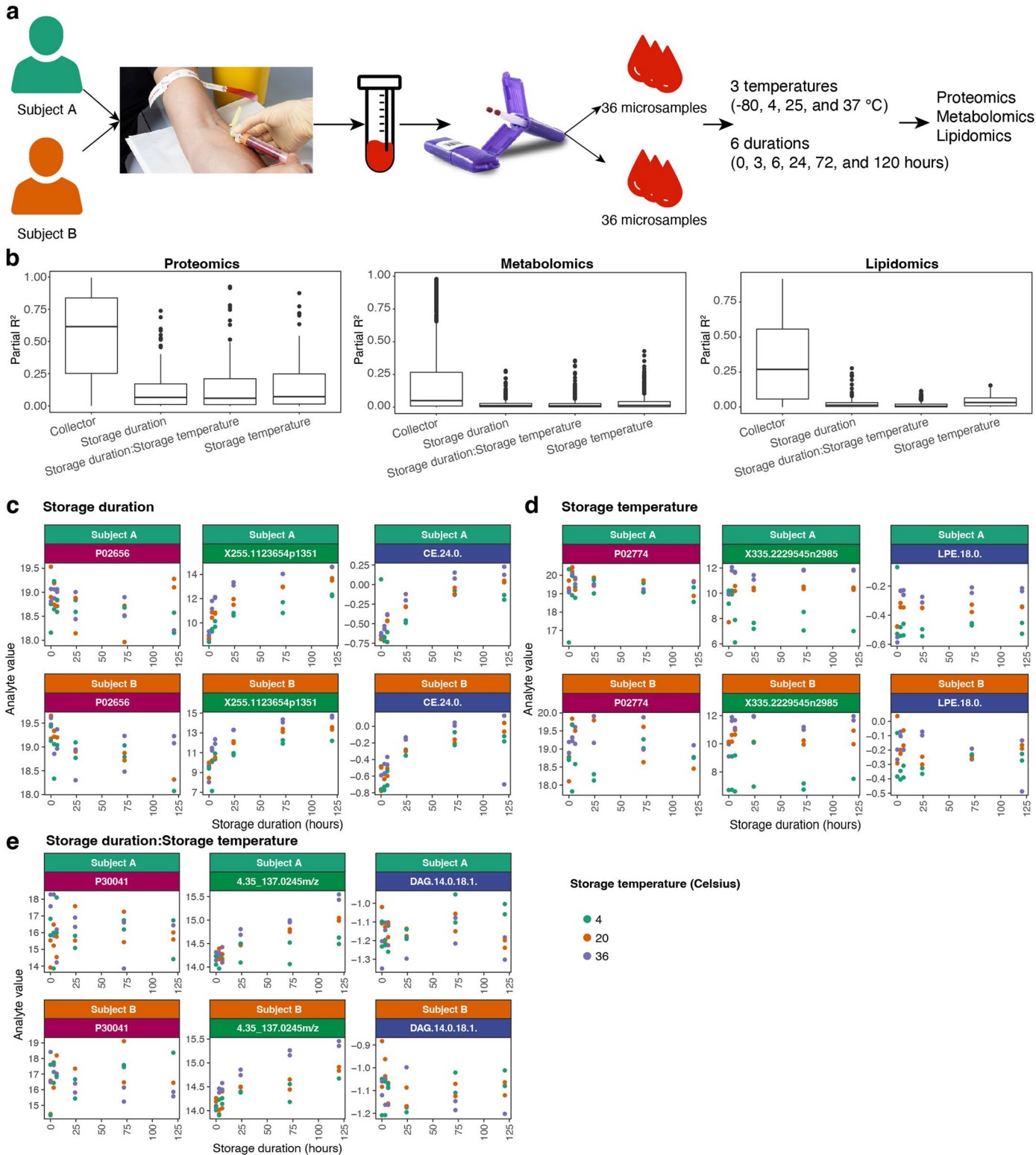

**Extended Data Fig. 1 | Stability analysis of the microsampling approach.**
**a,** The study design of the protein, metabolite, and lipid stability analyses in microsamples. **b,** The partial R² distribution for proteins, metabolites, and lipids. The most affected protein, metabolite, and lipid by storage duration (c), temperature (d) and interaction effect (e), respectively. The icons used in this figure are from iconfont.cn.

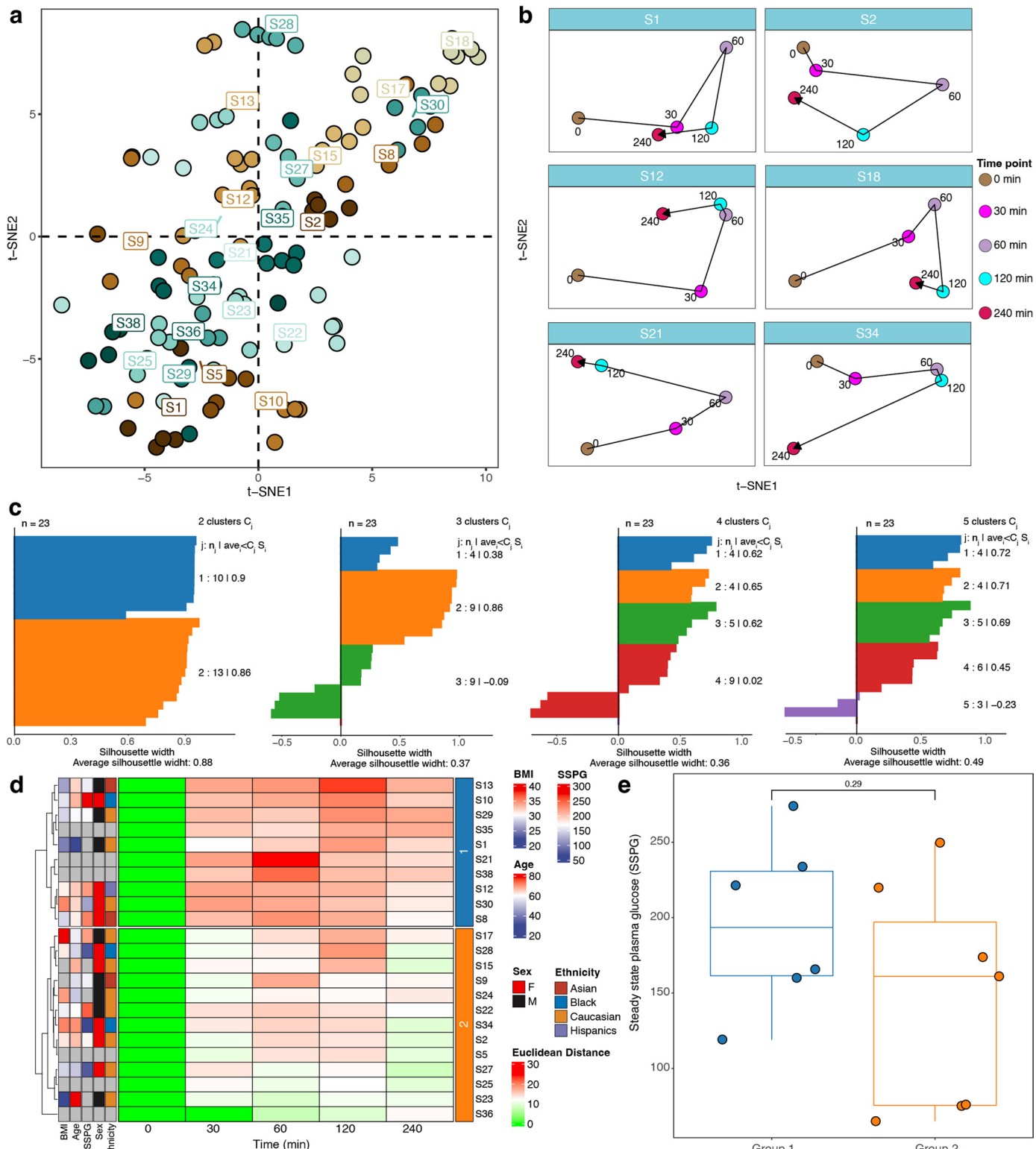

**Extended Data Fig. 2 | Metabolic phenotyping separates samples and subjects. a,** tSNA plot using all samples from all the participants. Colors represent the participants. **b**, tNSA plots for 6 participants. Colors represent the timepoints. The timepoints are also labeled on the plot. **c**, Silhouette plots for consensus clustering with group numbers 2, 3, 4, and 5. When the group number is 2, the Silhouette width achieves the highest value, so the group number is set as 2 for subsequent analysis. **d**, Heatmap plot showing differential clustering of molecular features in various samples compared to baseline (0 min) for each participant. Green represents low distance, and red represents high distance. **e**, The SSPG values for participants (only 13 participants) in group 1 and group 2.

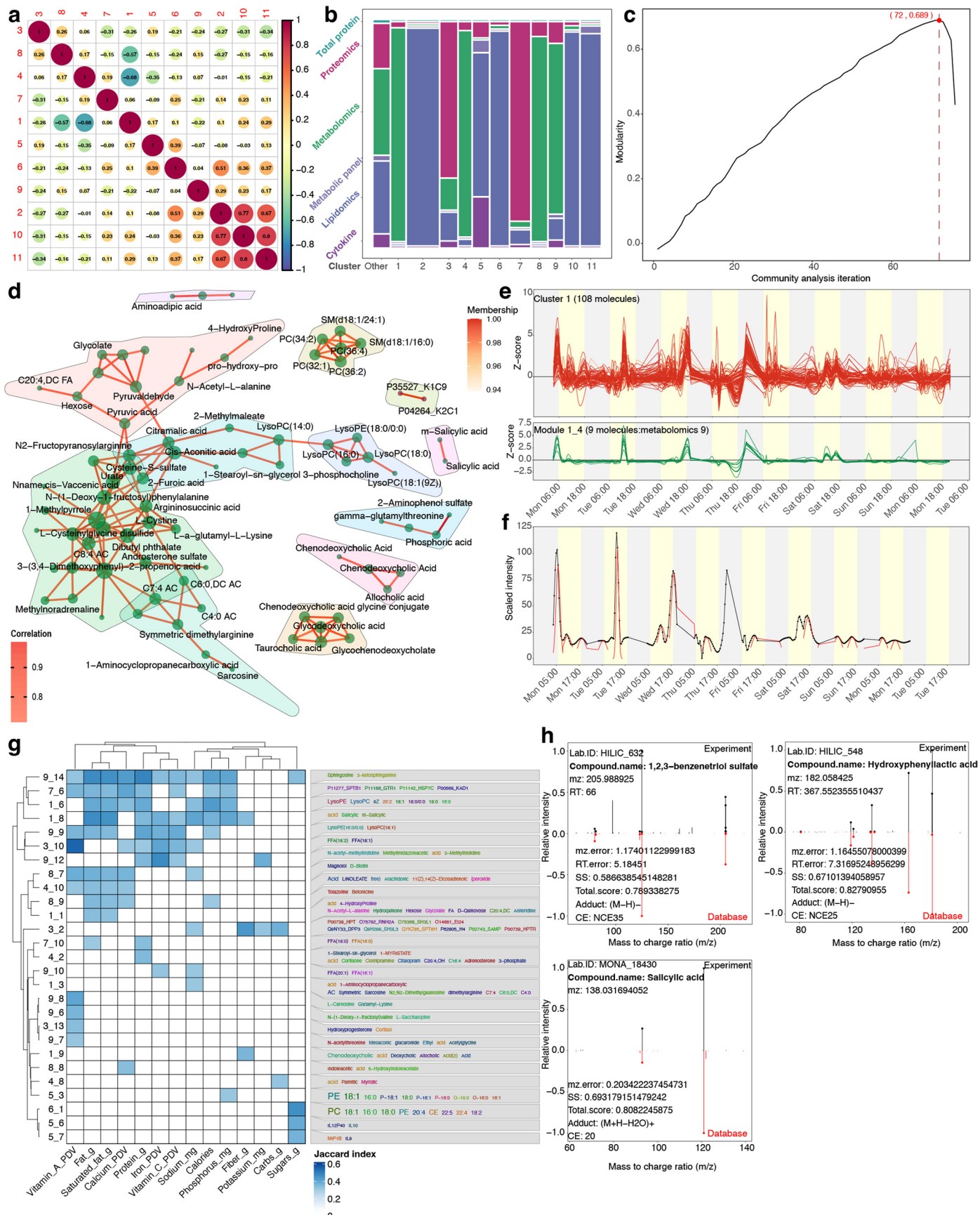

**Extended Data Fig. 3 | See next page for caption.**

**Extended Data Fig. 3 | Wearable and multi-omics data can reflect the individual's health status. a,** The Spearman correlations between all the clusters from all molecules in 24/7 study using the Fuzzy $c$-means clustering. **b**, The mosaic plot shows the molecules' classes for 11 clusters. **c**, The maximum modularity observed in our correlation network community analysis for cluster 1 was 0.689 at iteration 72 of community pruning. **d**, The molecule detection from the correlation network. The molecules have more connections inside than outside and are grouped as a module. **e**, Cluster 1 and Module 1_4 from it. **f**, Peak detection from the module using the peak detection algorithm. **g**, The heatmap to show the association between modules and nutrition. **h**, MS$^2$ spectra matching plots for 1,2,3-benzenetrlol sulfate, Hydroxyphenyllactic acid, and Salicylic acid, respectively.

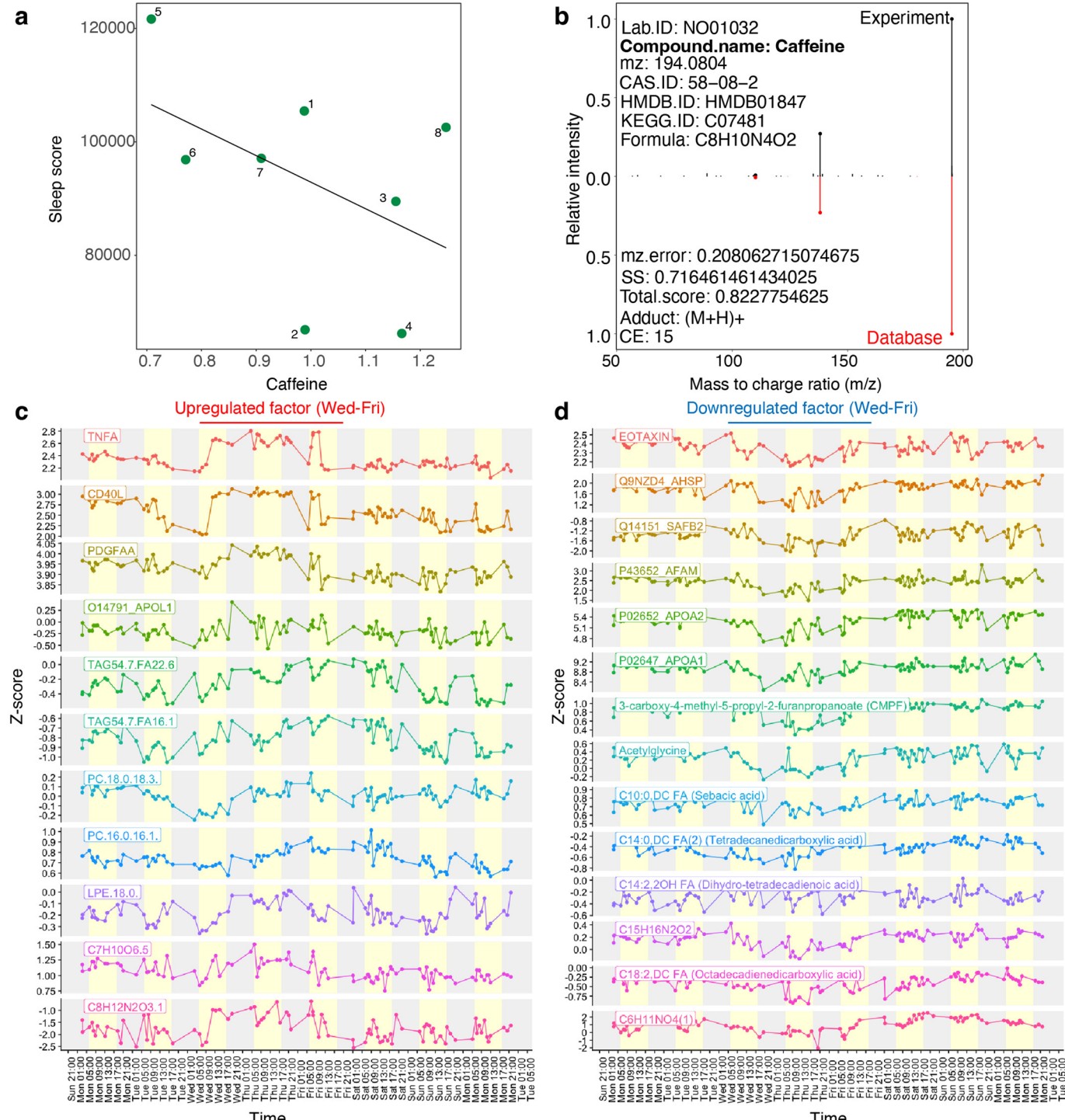

**Extended Data Fig. 4 | Wearable and multi-omics data can reflect the individual's health status. a,** The correlation plot between Caffeine intensity and sleep score. **b**, The MS$^2$ spectra matching plot for Caffeine. **c**, Molecules that were upregulated from Wednesday to Friday. **d**, Molecules that were downregulated from Wednesday to Friday.

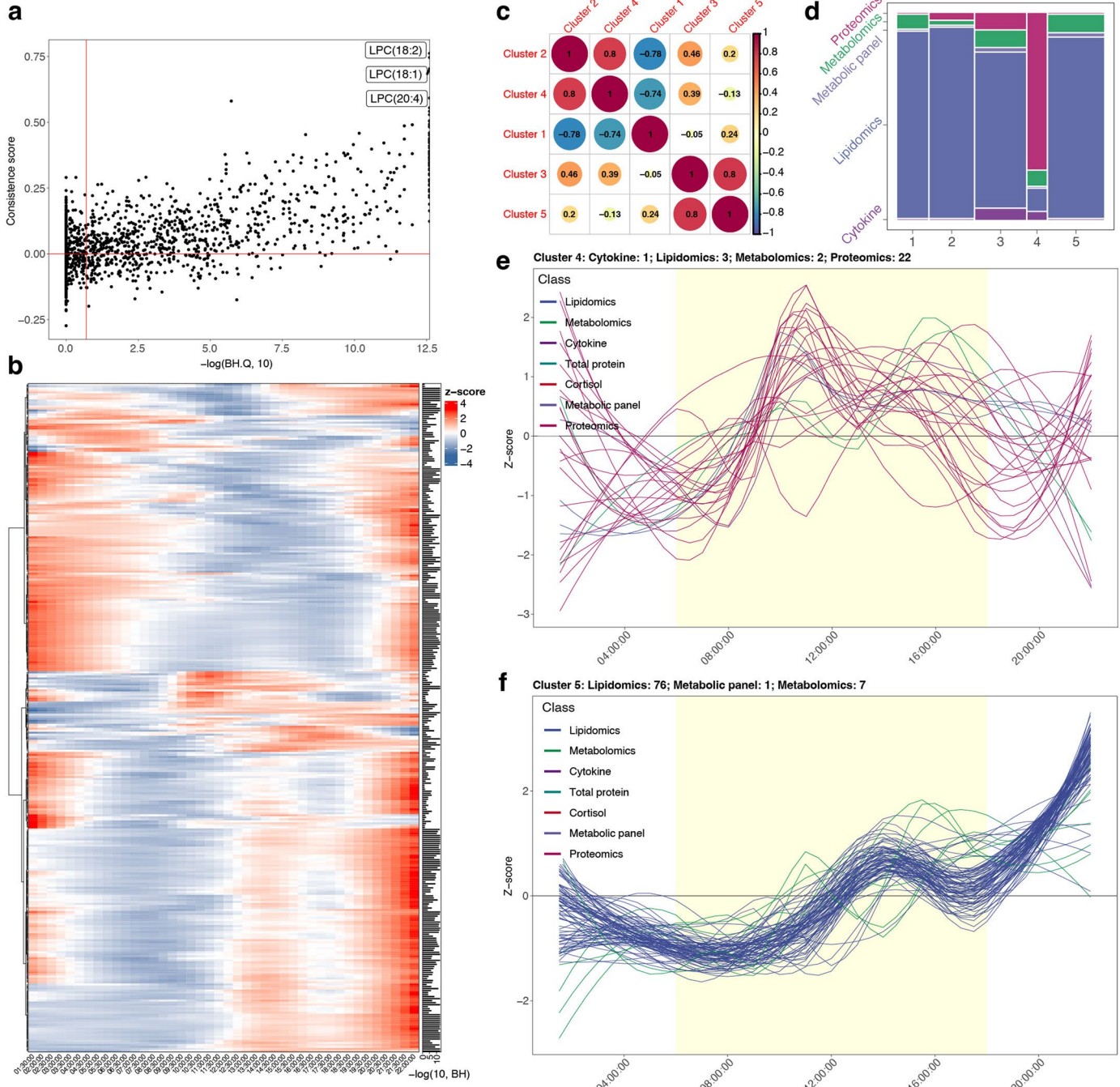

**Extended Data Fig. 5 | Circadian rhythm analysis for internal molecules.**
**a,** Consistence scores versus circadian rhythm p-values (-log10). **b,** Heatmap to show all the circadian molecules. **c,** Spearman correlation plot to show the correlations between 5 clusters from circadian molecules. **d,** The components of all 5 clusters. **e,** Cluster 4 contains 1 cytokine, 3 lipids, 2 metabolites, and 22 proteins. **f,** Cluster 3 contains 76 lipids, 1 metabolic panel, and 7 metabolites.

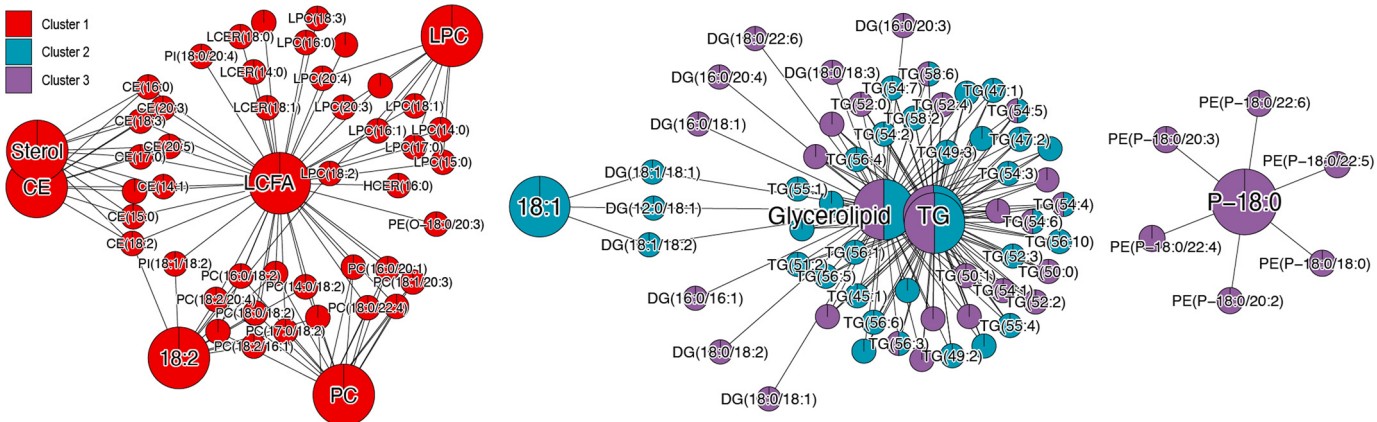

**Extended Data Fig. 6 | Lipid enrichment results for lipids in clusters 1–3.** Red represents cluster 1, dark green represents cluster 2 and purple represents cluster 3.

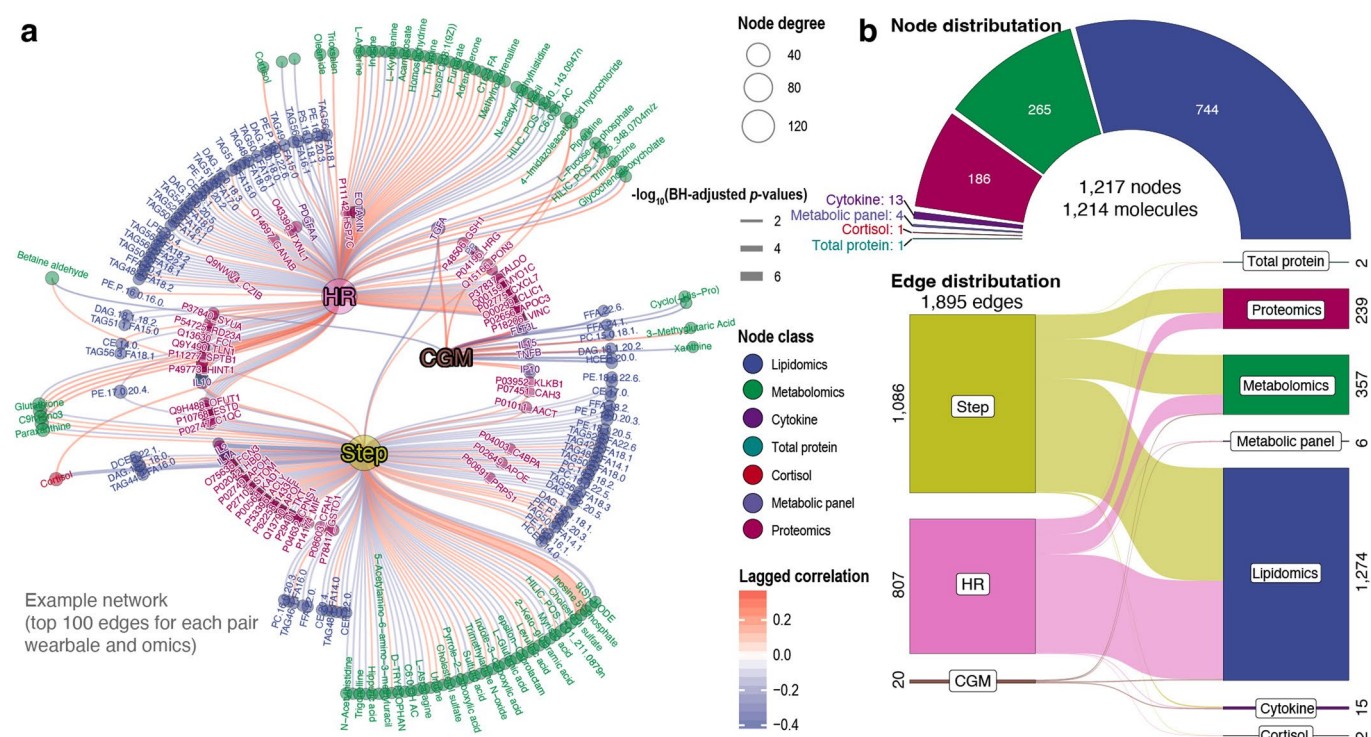

**Extended Data Fig. 7 | Wearable data and internal molecule causal association network. a,** Example association network between wearable data and internal molecules. **b**, Node and edge distribution of association network.

# Reporting Summary

## Statistics

For all statistical analyses, confirm that the following items are present in the figure legend, table legend, main text, or Methods section.

| n/a | Confirmed | |
|---|---|---|
| ☐ | ☒ | The exact sample size (*n*) for each experimental group/condition, given as a discrete number and unit of measurement |
| ☐ | ☒ | A statement on whether measurements were taken from distinct samples or whether the same sample was measured repeatedly |
| ☐ | ☒ | The statistical test(s) used AND whether they are one- or two-sided *Only common tests should be described solely by name; describe more complex techniques in the Methods section.* |
| ☐ | ☒ | A description of all covariates tested |
| ☐ | ☒ | A description of any assumptions or corrections, such as tests of normality and adjustment for multiple comparisons |
| ☐ | ☒ | A full description of the statistical parameters including central tendency (e.g. means) or other basic estimates (e.g. regression coefficient) AND variation (e.g. standard deviation) or associated estimates of uncertainty (e.g. confidence intervals) |
| ☐ | ☒ | For null hypothesis testing, the test statistic (e.g. *F*, *t*, *r*) with confidence intervals, effect sizes, degrees of freedom and *P* value noted *Give P values as exact values whenever suitable.* |
| ☒ | ☐ | For Bayesian analysis, information on the choice of priors and Markov chain Monte Carlo settings |
| ☒ | ☐ | For hierarchical and complex designs, identification of the appropriate level for tests and full reporting of outcomes |
| ☒ | ☐ | Estimates of effect sizes (e.g. Cohen's *d*, Pearson's *r*), indicating how they were calculated |

*Our web collection on statistics for biologists contains articles on many of the points above.*

## Software and code

Policy information about availability of computer code

| | |
|---|---|
| Data collection | ProteoWizard: Version. 3.0.19314-fb982f15b |
| Data analysis | R: 4.1.2; Rstudio: 2021.09.2. R package: colorspace_2.0-2  rjson_0.2.21  ellipsis_0.3.2 leaflet_2.1.0  rprojroot_2.0.2 circlize_0.4.14  GlobalOptions_0.1.2 clue_0.3-60  rstudioapi_0.13  mzR_2.28.0  affyio_1.64.0  fansi_1.0.2 xml2_1.3.3  codetools_0.2-18  ncdf4_1.19  doParallel_1.0.17  impute_1.68.0  knitr_1.37  jsonlite_1.7.3 cluster_2.1.2  vsn_3.62.0  png_0.1-7 readr_2.1.2  compiler_4.1.2  httr_1.4.2  assertthat_0.2.1 fastmap_1.1.0  lazyeval_0.2.2  limma_3.50.0  cli_3.2.0  htmltools_0.5.2  tools_4.1.2  gtable_0.3.0  glue_1.6.1 affy_1.72.0  dplyr_1.0.8  Biobase_2.54.0  cellranger_1.1.0  jquerylib_0.1.4  iterators_1.0.14  crosstalk_1.2.0  stringr_1.4.0 openxlsx_4.2.5  MSnbase_2.20.4  pcaMethods_1.86.0  hms_1.1.1  ProtGenerics_1.26.0 parallel_4.1.2 RColorBrewer_1.1-2 ComplexHeatmap_2.10.0 yaml_2.3.4  pbapply_1.5-0  yulab.utils_0.0.4  sass_0.4.0  stringi_1.7.6 highr_0.9  S4Vectors_0.32.3  foreach_1.5.2  BiocGenerics_0.40.0 zip_2.2.0  BiocParallel_1.28.3 shape_1.4.6 systemfonts_1.0.3  rlang_1.0.1  pkgconfig_2.0.3  matrixStats_0.61.0  mzID_1.32.0  evaluate_0.15  lattice_0.20-45 purrr_0.3.4  htmlwidgets_1.5.4  tidyselect_1.1.1  here_1.0.1  ggsci_2.9  plyr_1.8.6  bookdown_0.24  R6_2.5.1 IRanges_2.28.0  generics_0.1.2  DBI_1.1.2  pillar_1.7.0  withr_2.4.3  MsCoreUtils_1.6.0  tibble_3.1.6  crayon_1.5.0 utf8_1.2.2  plotly_4.10.0  tzdb_0.2.0  readxl_1.3.1  data.table_1.14.2  webshot_0.5.2 digest_0.6.29  tidyr_1.2.0  gridGraphics_0.5-1 ggplotify_0.1.0  bslib_0.3.0 |

For manuscripts utilizing custom algorithms or software that are central to the research but not yet described in published literature, software must be made available to editors and reviewers. We strongly encourage code deposition in a community repository (e.g. GitHub). See the Nature Portfolio guidelines for submitting code & software for further information.

## Data

Policy information about availability of data

All manuscripts must include a data availability statement. This statement should provide the following information, where applicable:

- Accession codes, unique identifiers, or web links for publicly available datasets
- A description of any restrictions on data availability
- For clinical datasets or third party data, please ensure that the statement adheres to our policy

All the data used in this study are provided as Supplementary Data.

## Human research participants

Policy information about studies involving human research participants and Sex and Gender in Research.

| | |
|---|---|
| Reporting on sex and gender | We have reported the sex information on the enrolled patients, which is determined by self-reporting |
| Population characteristics | Ensure shake study cohort: 21 out of 28 participants have completed demographic data. The median steady-state plasma glucose (SSPG) was 166, the median age was 64.2 years, and the median body mass index (BMI) was 29.7. Among all the participants, 38% were male, and 14.3% were Asian, 14.3% Black, 66.7% Caucasian and 4.8% Hispanic. |
| Recruitment | Ensure shake study cohort: Twenty-eight participants were enrolled in the Ensure shake study under an institutional review board (IRB)-approved protocol (IRB-47966 at Stanford University) with written consent. 21 out of 28 participants have completed demographic data.<br><br>24/7 study cohort: Only one participant (male, 64 years old) was enrolled in the 24/7 study under an IRB-approved protocol (IRB-23602 at Stanford University) with written consent. The participant was instructed to perform self-collected finger prick microsamples approximately every hour during waking hours and overnight periods sporadically for 7 days. |
| Ethics oversight | The study protocol was approved by the institutional review board at Stanford University (protocols IRB-47966 and IRB-23602). |

Note that full information on the approval of the study protocol must also be provided in the manuscript.

# Field-specific reporting

Please select the one below that is the best fit for your research. If you are not sure, read the appropriate sections before making your selection.

☒ Life sciences        ☐ Behavioural & social sciences        ☐ Ecological, evolutionary & environmental sciences

For a reference copy of the document with all sections, see nature.com/documents/nr-reporting-summary-flat.pdf

# Life sciences study design

All studies must disclose on these points even when the disclosure is negative.

| | |
|---|---|
| Sample size | No statistical methods were used to calculate the sample size. For the shake study, 28 participants were enrolled, and for the 24/7 study, 1 participant was enrolled. |
| Data exclusions | No data were excluded. |
| Replication | No replication was carried out. |
| Randomization | For all datasets, samples were assigned randomly to acquire omics data. |
| Blinding | The investigators were blinded to group information during data collection. At the time of sample acquisition and processing, the scientists were completely unaware of the sample group. |

# Reporting for specific materials, systems and methods

We require information from authors about some types of materials, experimental systems and methods used in many studies. Here, indicate whether each material, system or method listed is relevant to your study. If you are not sure if a list item applies to your research, read the appropriate section before selecting a response.

## Materials & experimental systems

| n/a | Involved in the study |
|-----|----------------------|
| ☒ | Antibodies |
| ☒ | Eukaryotic cell lines |
| ☒ | Palaeontology and archaeology |
| ☒ | Animals and other organisms |
| ☒ | Clinical data |
| ☒ | Dual use research of concern |

## Methods

| n/a | Involved in the study |
|-----|----------------------|
| ☒ | ChIP-seq |
| ☒ | Flow cytometry |
| ☒ | MRI-based neuroimaging |

