## [Peer Review File · Nature Biomedical Engineering]

Multi-omic microsampling for the profiling of lifestyle-associated changes in health

Corresponding author: Michael Snyder

Editorial note

This document includes relevant written communications between the manuscript's corresponding author and the editor and reviewers of the manuscript during peer review. It includes decision letters relaying any editorial points and peer-review reports, and the authors' replies to these (under 'Rebuttal' headings). The editorial decisions are signed by the manuscript's handling editor, yet the editorial team and ultimately the journal's Chief Editor share responsibility for all decisions.

Any relevant documents attached to the decision letters are referred to as **Appendix #**, and can be found appended to this document. Any information deemed confidential has been redacted or removed. Earlier versions of the manuscript are not published, yet the originally submitted version may be available as a preprint. Because of editorial edits and changes during peer review, the published title of the paper and the title mentioned in below correspondence may differ.

Correspondence

Tue 04 Oct 2022

Decision on Article nBME-22-2115

Dear Prof Snyder,

Thank you again for submitting to *Nature Biomedical Engineering* your manuscript, "Multi-omic microsampling captures health perturbations in a lifestyle context". The manuscript has been seen by three experts, whose reports you will find at the end of this message.

You will see that the reviewers appreciate the work. However, they express concerns about the apparently insufficient interpretation of the data, and provide useful suggestions for improvement, also regarding the presentation of the data. We hope that with significant further work you can address the criticisms and convince the reviewers of the merits of the study. In particular, we would expect that a revised version of the manuscript provides:

- * Further validation the microsampling methodology, as per the pertinent comments of all reviewers.
- * Analysis and Interpretation of the observed inter-individual variabilities in the metabolic-profiling study.
- * Biological insights into the intra-individual correlations seen in the whole-physiome-profiling study.
- * Discussion of the limitations of the methodology and of the overall multi-omic microsampling approach.
- * Discussion of the challenges towards real-world implementation of the methodology.
- * Thorough description of the study design and protocols, as per the various related questions from the reviewers.

When you are ready to resubmit your manuscript, please upload the revised files, a point-by-point rebuttal tothe comments from all reviewers, the reporting summary, and a cover letter that explains the main improvements included in the revision and responds to any points highlighted in this decision.

Please follow the following recommendations:

- * Clearly highlight any amendments to the text and figures to help the reviewers and editors find and understand the changes (yet keep in mind that excessive marking can hinder readability).
- * If you and your co-authors disagree with a criticism, provide the arguments to the reviewer (optionally, indicate the relevant points in the cover letter).
- * If a criticism or suggestion is not addressed, please indicate so in the rebuttal to the reviewer comments and explain the reason(s).
- * Consider including responses to any criticisms raised by more than one reviewer at the beginning of the rebuttal, in a section addressed to all reviewers.
- * The rebuttal should include the reviewer comments in point-by-point format (please note that we provide all reviewers will the reports as they appear at the end of this message).
- * Provide the rebuttal to the reviewer comments and the cover letter as separate files.

We hope that you will be able to resubmit the manuscript within 20 weeks from the receipt of this message. If this is the case, you will be protected against potential scooping. Otherwise, we will be happy to consider a revised manuscript as long as the significance of the work is not compromised by work published elsewhere or accepted for publication at *Nature Biomedical Engineering*.

We hope that you will find the referee reports helpful when revising the work. Please do not hesitate to contact me should you have any questions.

Best wishes,

Pep

Pep Pàmies
Chief Editor, Nature Biomedical Engineering

Reviewer #1 (Report for the authors (Required)):

The authors presented a new approach to perform frequent biomarker analysis using low blood volume. A commercial micro-sampler was used to collect 10 uL of blood every 30 or 60 minutes. Blood from the micro sampler was analyzed, following a protocol developed by the authors, for the measurement of thousands of metabolites, lipids, cytokines, and protein molecules. Variations of such molecules were observed after drinking a complex protein shake. Circadian rhythm correlation was found for some lipids, proteins, and metabolites. The authors present several statistical analyses for clustering and correlation analysis. The paper is well written, and the discussion is of high interest to the readers of *Nature Biomedical Engineering*. The authors should revise the manuscript to address the following comments.

1. For the food ingestion study, additional validation should be included to support the data obtained by microsampling. For example, performing measurements with well-known responses to validate the new microsampling methodology. Using a controlled simple mixture of proteins, metabolites, and lipids, the protocol can be validated and controls can be performed by alternating the concentration of the desired molecule.
2. Controlled targeted tests could be performed for the supposed molecules related to the drink ingestion

and the allegedly circadian rhythm-related molecules.

3. A summary table could be included in the supporting information with some representative molecules and their main discovered correlation (fast variation, strong correlation with specific pathway, correlation with companion molecule, etc.).

4. The error related to the proposed microsampling is of considerable extent (Fig2,e, f, g), please discuss the accuracy of the measurements compared with the dry blood spot approach.

5. Once the authors have demonstrated that some molecules can vary their concentrations as fast as 30 minutes, please discuss why 1 – 2 hour interval was used for the 24/7 study. The novelty claim of the proposed protocol is to be able to sample blood in small volumes and painlessly in a scale of minutes.

6. Please discuss what is the actual time window (defined by the sample preparation and analysis) for the user to receive their results after sending their samples to be analyzed.

7. Please discuss what would be the most accurate protocol when using data from a large study, once the individual profile differs greatly from the average. What are the limitations of each approach (personalized analysis vs group study)?

Some minor comments:

“Metabolites and lipids that were not correlated well (Spearman correlation < 0.5) were enriched for amino acids and triglycerides (TAGs), respectively (Figure S2c,d).” Please check, there is no figure S2d.

What are the units of “y” axis for figure 4c?

“Each participant collected one microsample (defined as 0 min),” typo, microsample.

“... summarize the pattern of changes associated with consumption time (Figure 3d, see Methods).” Figure 2d? Please check.

Reviewer #2 (Report for the authors (Required)):

In this paper by Shen, et al., the authors use a combination of multi-omic sampling from bloods and from wearables to monitor the status of patients at various time scales, and following specific perturbations such as the ingestion of a nutritional shake, the taking of aspirin or ingestion of coffee, or periodicity of a sleep/wake cycle. The paper is novel and interesting, and provides new insights into a host of biological functions in healthy individuals, including distinct circadian rhythms for different analytes, the connection between wearables data and clinical labs on time-scales not previously accessed. This paper should be an excellent candidate for NBME after some revisions.

Some of the results I found surprising and encouraging was the strong correlations between microsampled biospecimens and classical clinical assays, which is incredibly encouraging from the point of view of remote and/or high frequency monitoring.

My overall critique of the paper is that this paper may actually contain too much, and sometimes ‘less-is-more’ when teaching. Making it through the very dense figures, and the equally dense supplementary figures (18 of them!) is a bit of a slog. Part of the issue is that there are multiple studies reported here – some of which do need to be reported together, but this is still a lot. These include, the ensure shake study of a small group of participants, the circadian rhythm/high frequency study of one individual, the study to validate microsampling against clinical labs, and the study to establish relationships between wearables and molecular assays. Further, for most studies, very little biological interpretation is provided – i.e. there are lots of observations and lots of correlations and splitting of patients or analytes into different clusters, etc., but very little interpretation of what these splits mean, beyond a restating of the data. I’ll try to give a few examples below of what I mean.

Integrating Figures S6-S10 into a single chart that provides a more complete revelation of the molecular heterogeneity in the subjects would be very useful. As presented, the data simply shows heterogeneity, which is pretty much expected, and is not particularly useful as a tool to interpret the findings from this work. Replacing these 5 plots with a single plot that reveals a little more biology of that heterogeneity would be very helpful. i.e. – is it possible that some patients simply don’t have the enzymes to process certain metabolites, and so that skews the kinetics of their responses to the shake? I am not suggesting more

experiments, but perhaps these variations might be attributable to participant variations in the gut microbiome, and might be interpreted through a metabolic pathway (flux) analysis?

Fig S11. The negative correlation between SSPG and insulin sensitivity score (provided as the free fatty acid) is pretty tricky to pull out, as the text and the labels are incomplete. A brief description of why the free fatty acid level is equated with an insulin sensitivity score would be helpful to non-experts. There is a typo in the figure labels of part b. In terms of interpreting this individual-to-individual variability, other than insulin resistance, are there other metrics that one might look for to help interpretation of the data?

The circadian rhythms found within the one high frequency sampled individual are very interesting, and this is a section that could probably be readily expanded into its own paper, as it is very distinct from the Ensure shake study. Are they metabolic processes that are suggested by the differential rhythmic kinetics of the various metabolites? While the overall data set is very impressive, the analysis is largely correlations and clustering, and one is left a bit hanging in terms of any type of interpretation. With so much data, surely some interpretation is possible?

Finally, on the section on wearables vs blood analytics and causal relationships. The initial causal relationship between exercise and heart rate, the time lag, and how those metrics are mathematically established, is illustrative and intuitive. This intuition is bolstered by the well understood causal relationships between respiration/O₂, heart rate, and exercise. However, most of this section which appears to state that causal relationships can be extracted, is really less clear. In my mind, causal relationships require some level of mechanistic understanding with perturbations required to reveal the cause and effect. In other words, I can't really understand what the authors are calling as relationships determined from their data as 'causal,' and which ones are correlative. I don't really see any provencausal relationships that are uniquely resolved by this data set, but I might not understand the logic the authors are using. For sure, for causal relationships, the bar is obviously much higher relative to correlative associations.

Finally, something that is missing – is how might such a study be applied in a real world setting. When might such high density kinetic measurements provide real value. If that isn't clear, and the point of this paper is to just show that these kinds of measurements are possible, then ok. However, it would help readers to understand where this type of science and engineering is headed.

Reviewer #3 (Report for the authors (Required)):

The manuscript reports the use of the VAMS®, a sampling device capable of collecting micro-volumes of blood (10 ul), to monitor metabolites, lipids, proteins in blood samples from volunteers in a living setting. Two case studies are very (too) briefly presented. Both studies identified markers that changed within a 240 min time frame.

Such a multi -omics approach with strong bioinformatics and statistical analysis is very useful to get a comprehensive view of a complex biological event. This type of approach will probably become the standard for achieving a deep understanding of physiopathological events.

In my opinion, this manuscript presents a lot of results but without enough details and explanations. It is thus difficult to understand what the purpose of these studies is and what the take-home message is. Also, the ultimate value of these studies in the particular context of this lifestyle is unclear, especially at these sampling times. In addition, some major technical issues should be addressed (some examples are provided below). Very enthusiastic conclusions and perspectives are drawn without any mention of all the problems, artifacts and challenges that can be expected with this type of approach.

The main limitations of this study

- Identification of the rationale of the 2 cases studies (why those time courses? Provide more biological interpretation of the results in line with the identification and the correlation)
- Small number of individuals compared to the enormous inter-individual variability.
- Lack of validation of the targeted -omics approaches according Agency guidelines (accuracy, precision, matrix effects...)
- Lack of information concerning the protocols for marker identifications
- Poor design for the stability studies (i.e. stability study on 5 days meaning that all the samples should be

analytically processed within this time frame; is it really realistic?).

I would recommend focusing on a single case study that provides much more information on the rationale for the investigation, the biological significance of the results and the experimental conditions and their validation (cf international guidelines, as well as data from the QC samples).

Tue 22 Nov 2022

Decision on Article nBME-22-2115A

Dear Prof Snyder,

Thank you for your revised manuscript, "Multi-omic microsampling captures health perturbations in a lifestyle context". Having consulted with the original reviewers (whose brief comments you will find at the end of this message), I am pleased to write that we shall be happy to publish the manuscript in *Nature Biomedical Engineering*.

We will be performing detailed checks on your manuscript, and in due course will send you a checklist detailing our editorial and formatting requirements. You will need to follow these instructions before you upload the final manuscript files.

In the meantime, please do consider streamlining the main text and main figures so as to make the work easier to read and interpret. In particular, you can make use of Extended Data figures to better organize the data.

Best wishes,

Pep

Pep Pàmies
Chief Editor, Nature Biomedical Engineering

Reviewer #1 (Report for the authors (Required)):

The authors have fully addressed my comments.

Reviewer #2 (Report for the authors (Required)):

The authors have addressed most of my concerns, and I don't need to see the paper back, but the comment I made earlier that 'less is more' is something that very much applies to this manuscript. I hope that the editors and author can come together to figure out how to make this manuscript a little more digestible. It is fine work, and so I don't want to hold things back by another round of refereeing.

Reviewer #3 (Report for the authors (Required)):

The authors have made substantial improvements to the manuscript. The objectives, the discussion of the results, and the take-home message have been clarified, although the three studies have been retained and the content remains "heavy" to read. More biological interpretations have been introduced.

Of the changes to the manuscript, one seems incorrect to me: on p19, lines 2-4 "In addition, DBS requires a drying period before the sample can be sealed and shipped, which may also introduce variations in the analysis compared to VAMS." As far as I know, VAMS must also be dried for 2 hours before being safely transported.

Rebuttal 1

Response to the reviewers:

The authors thank the reviewers for their helpful comments and constructive suggestions. We feel these comments and suggestions have strengthened our manuscript considerably. We also would like to thank the positive feedback from all three reviewers. We have conducted new experiments and analyses and revised the manuscript to address the reviewers' comments and concerns. We have responded to all the comments point by point below in blue.

Reviewer #1

Remarks to the Author. The authors presented a new approach to perform frequent biomarker analysis using low blood volume. A commercial micro-sampler was used to collect 10 uL of blood every 30 or 60 minutes. Blood from the micro sampler was analyzed, following a protocol developed by the authors, for the measurement of thousands of metabolites, lipids, cytokines, and protein molecules. Variations of such molecules were observed after drinking a complex protein shake. Circadian rhythm correlation was found for some lipids, proteins, and metabolites. The authors present several statistical analyses for clustering and correlation analysis. The paper is well written, and the discussion is of high interest to the readers of Nature Biomedical Engineering. The authors should revise the manuscript to address the following comments.

Response: We thank the reviewer for the in-depth review of our manuscript. We have responded to the comments point by point below.

Comment #1. For the food ingestion study, additional validation should be included to support the data obtained by microsampling. For example, performing measurements with well-known responses to validate the new microsampling methodology. Using a controlled simple mixture of proteins, metabolites, and lipids, the protocol can be validated and controls can be performed by alternating the concentration of the desired molecule.

Response: We thank the reviewer for the constructive suggestions. We note that we did target specific lipids, cytokines, and hormones, which we were able to detect in our targeted assay. However, the reviewer's suggestion to perform measurements with a well-known mixture (such as a simple mixture of proteins, metabolites, or lipids) is a great idea to demonstrate and validate that microsampling could detect the real molecular signatures from the blood samples. Therefore, according to the reviewers' suggestion, we have now performed two experiments and analyses using controlled mixtures.

(1) We conducted an experiment to acquire the metabolomics data of Ensure shake in our study and compare it to the data acquired from the participant's blood microsamples. We felt this was the best validation as the shake is well characterized and is a complex mixture of compounds of the type described by the reviewer. We collected metabolomics data from Ensure shake using mass spectroscopy and processed and analyzed the data using the same protocol used by the participant samples in Ensure shake study. The compounds from Ensure shake were then matched with the data from the participants' microsamples. Nearly 50% of the compound found in the Ensure shake can be detected in the blood and most of the undetected metabolites were of lower abundance (**Supporting Figure 1**). Importantly, of 21 high-interest metabolites that changed in the blood upon ingestion, 17 are present in the Ensure shake. This result demonstrates that the microsampling approach could detect the molecular signatures from ingested samples, including high interest targets.

Supporting Figure 1. The overlap of compounds (metabolites) from Ensure shake and participants' microsamples in Ensure shake study.

(2) In the 24/7 study, we have the food logging of the participant and can compare general carbohydrate consumption with specific carbohydrates in the blood. After checking the food logging data, we found that the participant ate high Carbohydrate food at 11:45 am on 4/30/2019 and ate low Carbohydrate food the next day (5/1/2019, **Supporting Figure 2a**). So the microsamples between 11:45 am and 12:00 pm on 4/30/2019 were labeled as day 1, and microsamples between 11:45 am and 12:00 pm on 5/1/2019 were labeled as day 2. The mean values of the Carbohydrate food were 131.8 g and 31.9 g, respectively. Then the two carbohydrate metabolites (Fructose, Pyruvic acid) in microsamples from two days were extracted and shown as a boxplot (**Supporting Figure 2b**). The median values of Carbohydrate metabolites are 7.8 and 4.7, respectively. This result demonstrates that the omics data from microsamples roughly reflect the concentration of the food the participants consumed.

Supporting Figure 2. The omics data from microsamples can reflect the food intake. a, The food logging data (Carbohydrates) of the participant in the 24/7 study. **b**, The Carbohydrate metabolites (Fructose, Pyruvic acid) intensity distribution on two days.

In summary, these two additional experiments and analyses demonstrate that the omics data can detect the molecules from the food participants consumed. These results have been added to our revised manuscript (**page 6, lines 25-30; page 11, lines 20-25**).

Comment #2. Controlled targeted tests could be performed for the supposed molecules related to the drink ingestion and the allegedly circadian rhythm-related molecules.

Response: We thank the reviewer for the suggestion. We are not entirely sure what the reviewers are getting at. For targeted validation, please look at the response to comment #1, where we have now analyzed the shake metabolites. If the reviewer believes we have not performed validation of the annotated metabolites, please note that we have performed validation using standards for all of our known metabolites--this is now indicated in the methods (**page 23, lines 23-27**).

Comment #3. A summary table could be included in the supporting information with some representative molecules and their main discovered correlation (fast variation, strong correlation with specific pathway, correlation with companion molecule, etc.).

Response: We thank the reviewer for the suggestion. We have added a summary Table with representative molecules (amino acids in cluster 1, carbohydrates in cluster 1, acylcarnitines in cluster 3, and cytokines/hormones in clusters 1 and 3, *etc.*) as **Supplementary Table 5** in our revised manuscript.

Comment #4. The error related to the proposed microsampling is of considerable extent (Fig2,e, f, g). Please discuss the accuracy of the measurements compared with the dry blood spot approach.

Response: We thank the reviewer for the suggestion and apologize for the misunderstanding. The error bars in **Fig. 2e,f,g** are not the errors for the microsampling method but the variations from the different participants. In those figures, the x-axis shows samples collected at different time points, and the y-axis shows the normalized abundance of molecular features captured in those samples (participants). For the accuracy of the VAMS and dry blood spot (DBS), previous studies have shown that both VAMS and DBS show a similar good analytical performance for targeted compound and protein analysis (**Paniagua-González et al. 2022; Andersen et al. 2018**). However, for the DBS, the hematocrit effect (thickness of your blood) affects the resulting spot size, which can introduce variation in analysis. Additionally, DBS requires a drying period (up to 60 minutes) before the sample can be sealed and shipped, which may also introduce variations in the analysis compared to VAMS (**Spooner et al. 2015**). This discussion has now been added to the revised manuscript (**page 18, line 13; page 19, lines 1-4**).

References

Andersen, Ida Kristine Lysgaard, Cecilie Rosting, Astrid Gjelstad, and Trine Grønhaug Halvorsen. 2018. "Volumetric Absorptive MicroSampling vs. Other Blood Sampling Materials in LC-MS-Based Protein Analysis - Preliminary Investigations." *Journal of Pharmaceutical and Biomedical Analysis* 156 (July): 239–46.

Paniagua-González, Lucía, Elena Lendoiro, Esteban Otero-Antón, Manuel López-Rivadulla, Ana de-Castro-Ríos, and Angelines Cruz. 2022. "Comparison of Conventional Dried Blood Spots and Volumetric Absorptive Microsampling for Tacrolimus and Mycophenolic Acid Determination." *Journal of Pharmaceutical and Biomedical Analysis* 208 (January): 114443.

Spooner, Neil, Philip Denniff, Luc Michielsen, Ronald De Vries, Qin C. Ji, Mark E. Arnold, Karen Woods, et al. 2015. "A Device for Dried Blood Microsampling in Quantitative Bioanalysis: Overcoming the Issues Associated Blood Hematocrit." *Bioanalysis* 7 (6): 653–59.

Comment #5. Once the authors have demonstrated that some molecules can vary their concentrations as fast as 30 minutes, please discuss why 1–2 hour interval was used for the 24/7 study. The novelty claim of the proposed protocol is to be able to sample blood in small volumes and painlessly in a scale of minutes.

Response: We thank the reviewer for this excellent comment and now make this more precise. We have revised the text to say that this method enables frequent sampling on the order of minutes or hours and the collection of a large number of samples from a single individual. In addition, some of the samplings in the 24/7 study were performed in 30 min (before and after exercise, **Figure S8b**). So we have revised the text here as well. Please note that from a practical standpoint, collecting ~every hour during most of the waking period is a considerable effort (nearly 100 samples!). We hope the reviewer is satisfied with the more precise text (**page 11, lines 1-6; page 26, lines 14-27**).

Comment #6. Please discuss what is the actual time window (defined by the sample preparation and analysis) for the user to receive their results after sending their samples to be analyzed.

Response: We thank the reviewer for the suggestion. In our study, all the microsamples were stored at -80 °C after sample collection. For the Ensure shake study, they were returned in one day or less by FedEx. For the 24/7 study, they were frozen on dry ice immediately after collection. Then after all the microsamples were collected, we prepared and ran all the samples together as a batch to avoid batch effects. In the future, when we apply the microsampling method to the real world, the microsamples collected can potentially be run within days after collection. Note that for the two case studies in our manuscript, the users can receive their results within two days after sending their samples to the laboratory for analysis. We have added these points to our revised manuscript (**page 20, lines 20-21**).

Comment #7. Please discuss what would be the most accurate protocol when using data from a large study, once the individual profile differs greatly from the average. What are the limitations of each approach (personalized analysis vs group study)?

Response: We thank the reviewer for the constructive suggestion. For a group study that uses data from a large study, and the individual profile differs greatly from the average, the first thing we do to perform “data sanity checks” to ensure that there is not any systematic variation or batch effect. On the cleaned data, we run outlier detection. We identify individuals who fall outside the overall pattern and then attempt to investigate the underlying reason for the deviation-medical conditions, medications, or lifestyle abnormalities. Group analysis is usually performed to find the overall trend. In addition, when we perform group data analysis, we usually control for confounders (e.g. sex, age, BMI, *etc.*) and make adjustments to find the real and expected biological variation.

For the group study, as the reviewer mentioned, the individual profile may differ greatly from the average, which means that the overview conclusions from the whole cohort may not work on individuals--we have published this in several studies, including our recent fiber and exercise studies (Contrepois et al. 2020, Lancaster et al. 2022). Thus for the personalized analysis, the conclusion from the individual may not be extended to other people. This discussion has been added to our revised manuscript (**page 21, lines 6-16**).

References

Contrepois, Kévin, Si Wu, Kegan J. Moneghetti, Daniel Hornburg, Sara Ahadi, Ming-Shian Tsai, Ahmed A. Metwally, et al. 2020. “Molecular Choreography of Acute Exercise.” *Cell* 181 (5): 1112–30.e16.

Lancaster, Samuel M., Brittany Lee-McMullen, Charles Wilbur Abbott, Jeniffer V. Quijada, Daniel Hornburg, Heyjun Park, Dalia Perelman, et al. 2022. “Global, Distinctive, and Personal Changes in Molecular and Microbial Profiles by Specific Fibers in Humans.” *Cell Host & Microbe* 30 (6): 848–62.e7.

Some minor comments:

Comment #8. “Metabolites and lipids that were not correlated well (Spearman correlation < 0.5) were enriched for amino acids and triglycerides (TAGs), respectively (Figure S2c,d).” Please check, there is no figure S2d.

Response: We apologize for the typo and confusion. This typo (“**Figure S2c,d**”) has been corrected in our revised manuscript (“**Figure S2b,c**”).

Comment #9. What are the units of “y” axel for figure 4c?

Response: We apologize for the missed information. The unit for food is “gram”; the unit for step count is “Steps/minute”; the unit for heart rate is “Beats/minute”; the unit for CGM is “mg/dL”. These have been added in the revised **Figure 4c**.

Comment #10. “Each participant collected one microsmaple (defined as 0 min),” typo, microsmaple.

Response: We apologize for the typo. This typo (“microsmaple”) has been corrected in our revised manuscript (“microsample”).

Comment #11. “.... summarize the pattern of changes associated with consumption time (Figure 3d, see Methods).” Figure 2d? Please check.

Response: We apologize for the typo. This typo (“**Figure 3d**”) has been corrected in our revised manuscript (“**Figure 2d**”).

Reviewer #2

Remarks to the Author. In this paper by Shen, et al., the authors use a combination of multi-omic sampling from bloods and from wearables to monitor the status of patients at various time scales and following specific perturbations such as the ingestion of a nutritional shake, the taking of aspirin or ingestion of coffee, or periodicity of a sleep/wake cycle. The paper is novel and interesting and provides new insights into a host of biological functions in healthy individuals, including distinct circadian rhythms for different analytes, the connection between wearables data and clinical labs on time-scales not previously accessed. This paper should be an excellent candidate for NBME after some revisions.

Some of the results I found surprising and encouraging was the strong correlations between microsampled biospecimens and classical clinical assays, which is incredibly encouraging from the point of view of remote and/or high-frequency monitoring.

My overall critique of the paper is that this paper may actually contain too much, and sometimes ‘less-is-more’ when teaching. Making it through the very dense figures, and the equally dense

supplementary figures (18 of them!) is a bit of a slog. Part of the issue is that there are multiple studies reported here – some of which do need to be reported together, but this is still a lot. These include, the ensure shake study of a small group of participants, the circadian rhythm/high frequency study of one individual, the study to validate microsampling against clinical labs, and the study to establish relationships between wearables and molecular assays. Further, for most studies, very little biological interpretation is provided – i.e. there are lots of observations and lots of correlations and splitting of patients or analytes into different clusters, etc., but very little interpretation of what these splits mean, beyond a restating of the data. I'll try to give a few examples below of what I mean.

Response: We thank the reviewer for the in-depth review. This study's main aim is to develop the multi-omic microsampling approach, and two pilot case studies were only utilized to demonstrate its power and promising application in precision medicine. We did debate as to whether to break the two studies into separate papers, but we felt the overall paper would be better if everything was combined into a single manuscript. Unfortunately, this does leave less space for biological interpretations. However, we note that several studies using the microsampling approach are going on in our lab addressing specific biological questions where these can be covered in more detail. We also provided more biological interpretations according to the reviewer's suggestion below. Finally, we have tried to make the paper a bit less dense to help with the reading. We are happy to make our slides available to anyone who wants them. We have responded to the reviewer's comments point by point below.

Comment #1. Integrating Figures S6-S10 into a single chart that provides a more complete revelation of the molecular heterogeneity in the subjects would be very useful. As presented, the data simply shows heterogeneity, which is pretty much expected and is not particularly useful as a tool to interpret the findings from this work. Replacing these 5 plots with a single plot that reveals a little more biology of that heterogeneity would be very helpful. i.e. – is it possible that some patients simply don't have the enzymes to process certain metabolites, and so that skews the kinetics of their responses to the shake? I am not suggesting more experiments, but perhaps these variations might be attributable to participant variations in the gut microbiome, and might be interpreted through a metabolic pathway (flux) analysis?

Response: We thank the reviewer for the comments and suggestions. According to the reviewer's suggestion, **Figures S6-S10** have been integrated into a single figure (revised **Figure S6**) in our revised supplementary. The heterogeneity of participants is quantitatively measured by the metabolic score in our manuscript (**page 8, lines 16-18**). We agree that the heterogeneity of the participants' response to Ensure shake may be caused by the different enzymes or gut microbes in different participants. This can be explored to define the underlying mechanism by more omics data (microbiome) or new experiments (such as metabolic flux analysis) in the future. According to the reviewer's suggestion, his discussion has been added to our revised manuscript (**page 9, lines 16-19**).

Comment #2. Fig S11. The negative correlation between SSPG and insulin sensitivity score (provided as the free fatty acid) is pretty tricky to pull out, as the text and the labels are incomplete. A brief description of why the free fatty acid level is equated with an insulin sensitivity score would be helpful to non-experts. There is a typo in the figure labels of part b. In terms of interpreting this individual-to-individual variability, other than insulin resistance, are there other metrics that one might look for to help interpretation of the data?

Response: We thank the reviewer for the comments and suggestions. Some studies have demonstrated that elevated plasma levels of free fatty acids are associated with insulin resistance (Xin et al. 2019). SSPG (steady-state plasma glucose) is inversely related to insulin sensitivity (SSPG is higher in insulin-resistant subjects and lower in insulin-sensitive subjects). So the negative correlation between SSPG and insulin sensitivity score (calculated using the free fatty acid) is expected. According to the reviewer's suggestion, this brief description has been added to the revised manuscript (**page 9, lines 1-9**) and supplementary. In addition, to make this figure clearer for the readers, the names have been changed to the corresponding scores in revised **Figure S7b**.

We apologize for the typo, which has been corrected in our revised **Figure S7b**.

For interpreting individual-to-individual variability, other than insulin sensitivity, we utilized another five metrics to measure the individual's response to the Ensure shake, namely 1) carbohydrate, 2) lipid, 3) protein metabolism, 4) insulin secretion, and 5) inflammatory response (cytokines). We found that individuals varied significantly in their response to the Ensure shake for each of the metrics (**Fig. 3d**). This is mentioned in our manuscript, and we have modified it to make it clearer for the readers (**page 8, lines 16-21**).

References

Xin, Yanlu, Yunyang Wang, Jingwei Chi, Xvhua Zhu, Hui Zhao, Shihua Zhao, and Yangang Wang. 2019. "Elevated Free Fatty Acid Level Is Associated with Insulin-Resistant State in Nondiabetic Chinese People." *Diabetes, Metabolic Syndrome and Obesity: Targets and Therapy* 12 (January): 139–47.

Comment #3. The circadian rhythms found within the one high-frequency sampled individual are very interesting, and this is a section that could probably be readily expanded into its own paper, as it is very distinct from the Ensure shake study. Are they metabolic processes that are suggested by the differential rhythmic kinetics of the various metabolites? While the overall data set is very impressive, the analysis is largely correlations and clustering, and one is left a bit hanging in terms of any type of interpretation. With so much data, surely some interpretation is possible?

Response: We thank the reviewer for the positive feedback, pointing out that the circadian rhythm results are very interesting. We agree that the 24/7 study is an interesting and valuable dataset for the comprehensive physiome, which could be expanded into a single paper. However, it can be difficult to publish a paper on a single individual, and although we considered splitting this into two papers when we first prepared it, in the end, we felt that combining both the Ensure shake and 24/7 study into a single paper strongly demonstrated the application of the multi-omic microsampling approach. Additionally, all the datasets of the 24/7 study have been provided and are public, so other people interested in the related studies can explore these data for their research.

In the revised paper, we now give more biological interpretations of the rhythmic lipids (**Figure 5**). LPC, PC, Sterol, and Cholesterol ester (CE) groups were significantly enriched in cluster 1. Previous work has shown that LPC and PC have circadian rhythm with peak concentrations in the evening, which is consistent with our result (Chua et al. 2013). For cluster 2, TG and Glycerolipid were significantly enriched, and for cluster 3, PE was significantly enriched (Figure 5f). Those different classes of lipids exhibit distinct circadian patterns. To explore if the circadian lipids were affected by the food intake, we then checked the

food logging data. We found that the fat nutrition intakes differed across eight days, meaning that the circadian lipids are not driven by the food intake. So the circadian lipids were driven by the individual rhythmic kinetics or gut microbes, which should be further studied by more experiments. This novel result has been added to our revised manuscript (**page 14, lines 25-36**).

Additionally, we now provide new several new biological interpretations and applications for the 24/7 study, which have been added to our revised manuscript (**page 17, lines 3-14**).

References

Chua, Eric Chern-Pin, Guanghou Shui, Ivan Tian-Guang Lee, Pauline Lau, Luuan-Chin Tan, Sing-Chen Yeo, Buu Duyen Lam, et al. 2013. “Extensive Diversity in Circadian Regulation of Plasma Lipids and Evidence for Different Circadian Metabolic Phenotypes in Humans.” *Proceedings of the National Academy of Sciences of the United States of America* 110 (35): 14468–73.

Comment #4. Finally, on the section on wearables vs blood analytics and causal relationships. The initial causal relationship between exercise and heart rate, the time lag, and how those metrics are mathematically established, is illustrative and intuitive. This intuition is bolstered by the well understood causal relationships between respiration/O₂, heart rate, and exercise. However, most of this section which appears to state that causal relationships can be extracted is really less clear. In my mind, causal relationships require some level of mechanistic understanding with perturbations required to reveal the cause and effect. In other words, I can't really understand what the authors are calling as relationships determined from their data as 'causal,' and which ones are correlative. I don't really see any proven causal relationships that are uniquely resolved by this data set, but I might not understand the logic the authors are using. For sure, for causal relationships, the bar is obviously much higher relative to correlative associations.

Response: We thank the reviewer for the comment and apologize for the confusion. We agree that the bar of the causal relationships is much higher than the correlation associations. However, when analyzing kinetics, events that precede other events are generally more likely to be causal and are often interpreted that way. Nonetheless, in the revised manuscript, we have added the phrase “potentially” causal relationships to be more careful. We also may it clear that all potential causal relationships that we found should be validated by more experiments in the future.

In addition, to be clearer, we have also updated the manuscript to clarify how we use the laggedCor (lagged correlation) method in the 24/7 study to extract the potential causal relationships between wearable and internal omics data (**page 30, lines 27-34**). Let's assume that X is wearable data and Y is internal omics data. In a real biological system, if X and Y have a causal relationship (X causes Y), Y often responds to X after a certain lapse of time. Such a lapse of time is called a lag time (**Supporting Figure 2**). So it means that X and Y change asynchronously (as **Supporting Figure 2** shows. Red line: X, black line: Y). To explore if X and Y have a potential causal relationship, we just shift the lag time between X and Y for matching and then calculate the correlation between them. Suppose the X and Y have a potential causal relationship and the lag time is T. In that case, we obtain the significantly highest correlation between X and Y with the shift time T. Like the example shown in revised **Fig. S14a**, we used the step count and heart rate as an example. We get the highest correlation between them with a lag time (-1min), which means that the step increases the heart rate and the lag time is around 1 min. This makes sense and shows that the

lagged correlation method could capture the potential causal relationships between two time-series data. All these points and data have also been added to our revised manuscript.

Supporting Figure 2. The illustration of lagged correlation.

Comment #5. Finally, something that is missing – is how might such a study be applied in a real-world setting. When might such high-density kinetic measurements provide real value? If that isn't clear, and the point of this paper is just to show that these kinds of measurements are possible, then ok. However, it would help readers to understand where this type of science and engineering is headed.

Response: We thank the reviewer for the suggestions. Because the multi-omic microsampling approach is fully remote, scalable, and highly temporal, it has potential application in the following fields in the near future. (1) Longitudinal biomarker discovery. The multi-omics microsampling is simple and not painful compared to the traditional blood collection method. Thus, it allows anyone to self-collect high-frequent (~ hourly) and high-quality blood microsamples anywhere for longitudinal biomarker discovery. (2) Personalized health monitoring. The people can collect blood samples at home without any help and then send the samples to the laboratory for data acquisition and analysis. If a significant abnormality is detected, the result is sent immediately to a physician. The physician would then be able to respond quickly with an intervention. (3) Therapeutic drug monitoring. The patients could collect microsamples frequently and remotely to monitor the drug-related compounds or biomarkers in the blood at a known time to guide dosage and result in optimized therapy. These points have been added as a paragraph to the discussion of our revised manuscript (**page 20, lines 27-41**).

Reviewer #3 (Report for the authors (Required)):

Remarks to the Author. The manuscript reports the use of the VAMS®, a sampling device capable of collecting micro-volumes of blood (10 ul), to monitor metabolites, lipids, proteins in blood samples from volunteers in a living setting. Two case studies are very (too) briefly presented. Both studies identified markers that changed within a 240 min time frame.

Such a multi-omics approach with strong bioinformatics and statistical analysis is very useful to get a comprehensive view of a complex biological event. This type of approach will probably become the standard for achieving a deep understanding of physiopathological events.

In my opinion, this manuscript presents a lot of results but without enough details and explanations. It is thus difficult to understand what the purpose of these studies is and what the take-home message is. Also, the ultimate value of these studies in the particular context of this lifestyle is unclear, especially at these sampling times. In addition, some major technical issues should be addressed (some examples are provided below). Very enthusiastic conclusions and perspectives are drawn without any mention of all the problems, artifacts and challenges that can be expected with this type of approach.

Response: We thank the reviewer for the in-depth review. The main aim of this study is to develop the multi-omic microsampling approach, and two pilot case studies were only utilized to demonstrate its power and promising application in precision medicine. Additionally, according to the reviewers' suggestions, we have tried to make this clearer, and we have provided more biological interpretation for the 24/7 study (case study #2), which has been added to our revised manuscript (see below). We have also added the limitations and challenges in the revised manuscript (**page 17, lines 29-30; page 20, line 44; page 21, lines 1-2**). We have responded to all the comments point by point below.

The main limitations of this study:

Comment #1. Identification of the rationale of the 2 cases studies (why those time courses? Provide more biological interpretation of the results in line with the identification and the correlation)

Response: We thank the reviewer for the comments and constructive suggestions. In this manuscript, we focus on two powerful capabilities of microsampling: 1) to measure a comprehensive and complex dynamic response to a defined perturbation and reveal inter-individual variation, and 2) through high-frequency sampling over multiple days to reveal correlations between physiological parameters and molecular changes in an individual. As a proof of concept study, we aim to demonstrate new technological capabilities in this manuscript. We have now provided more rationale for the two studies in the revised manuscript. Moreover, although the very deep biological interpretation is not the main aim of the study, we have tried to improve it in the revised manuscript.

Specifically, according to the reviewer's suggestions, we have analyzed the data further and provided more biological interpretation for the 24/7 study in the revised manuscript. Pathway enrichment analysis was performed for proteins, metabolites, and lipids to achieve an overview of the omics data correlated with wearable data. Interestingly, the immunity-related pathways were enriched in the proteins negatively correlated with CGM signals in this individual (**Figure S14e**), which was not expected. This demonstrates the importance of following these responses at the individual level. As expected, we also found that glucagon signaling, oxidative phosphorylation pathways, and free fatty acids positively correlate with CGM (**Figure S14f,g**). Glucagon can raise the concentration of glucose and fatty acids in the bloodstream (Voet and Voet 2004), and oxidative phosphorylation can oxidize nutrients to release chemical energy. We found that the caffeine metabolism pathway positively correlates with heart rate (**Figure S14h**), consistent with previous studies (Bitar, Mastouri, and Kreutz 2015). We also found that the blood coagulation pathway positively correlates with heart rate (**Figure S14i**), and the neutrophil degranulation pathway negatively correlates with heart rate (**Figure S14j**). To our best knowledge, these associations provide new biological insights and hypotheses that should be validated in the future. Overview, these results demonstrate that the wearable can reflect the physiological status of the participant (**page 17, lines 3-14**).

References

Bitar, Abbas, Ronald Mastouri, and Rolf P. Kreutz. 2015. "Caffeine Consumption and Heart Rate and Blood Pressure Response to Regadenoson." *PLoS One* 10 (6): e0130487.

Voet, Donald, and Judith G. Voet. 2004. *Biochemistry*. John Wiley & Sons Incorporated.

Revised Supplementary Figure 14e. Pathway enrichment results for proteins negatively correlated with CGM. **f**, Pathway enrichment results for metabolites positively correlated with CGM. **g**, Lipid class enrichment results for lipids that positively correlated with CGM. **h**, Pathway enrichment results for metabolites positively correlated with heart rate (HR). **i**, Pathway enrichment results for proteins positively correlated with heart rate (HR). **j**, Pathway enrichment results for proteins negatively correlated with heart rate (HR).

Comment #2. Small number of individuals compared to the enormous inter-individual variability.

Response: We thank the reviewer for the comment. That is part of the point. Our dense sampling enables details analyses of individual responses. For the Ensure shake study, our study demonstrates the range and heterogeneity of the responses at individual levels. In the 24/7 study, we asked the participant to collect microsamples almost hourly (1-2 hour intervals) for 24 hours and 7 days, which is extremely difficult for traditional intravenous sampling but can be done with our approach. Additionally, the 24/7 study aims to monitor the individual health status longitudinally, so inter-individual variability is not an important issue for this study. To clarify the study's aims, we have added this to our revised manuscript, which we hope improves the rationale (**page 11, lines 1-6**).

Comment #3. Lack of validation of the targeted-omics approaches according to Agency guidelines (accuracy, precision, matrix effects...)

Response: Thanks for the reviewer's comment. This research study aims to measure relative changes over time in each participant. Developing a clinical diagnostic requires additional validation steps that the reviewer suggests. However, ours is not a clinical test but rather an approach that we present. We have added the reviewer's point to our revised manuscript to make it clearer for the readers (**page 20, lines 39-41**).

Comment #4. Lack of information concerning the protocols for marker identifications

Response: We thank the reviewer for the comments and apologize for the lack of detailed information about the protocols for marker identifications. The biomarker identification is only to identify differentially expressed molecules after consuming Ensure shake (case study #1). The protocol can be found on **page 27, lines 7-13**. Briefly, the time point 0 was set as the baseline, and all the other 4 time points were compared to the baseline to get the differentially expressed molecules. The paired Wilcoxon rank-sum test was utilized to get the *p*-values. The multiple comparisons were adjusted using the Benjamini & Hochberg (BH) method. The adjusted *p*-values less than 0.05 were considered as significantly differentially expressed molecules. This detailed protocol and other methods have been updated to make it clearer for readers.

Comment #5. Poor design for the stability studies (i.e. stability study on 5 days meaning that all the samples should be analytically processed within this time frame; is it really realistic?).

Response: We thank the reviewer for the comment and apologize for the misunderstanding. All the microsamples were stored at -80 °C after collection before they were processed and analyzed. We are making the assumption that there is little analyte degradation at -80 °C for blood samples which we have found previously, so this is reasonable. Thus, the stability analysis was designed to explore if the molecules from the microsamples are stable in different storage conditions (temperature and duration time) before they are stored at -80 °C. Due to the time- and temperature-sensitivity of some of the molecules, in the real-world use of the microsampling method, we would not let the microsamples sit at ambient storage (for example, room temperature) for more than 5 days. In fact, we would carry out rapid shipping or maybe even shipping at cold temperatures to remove the effects of storage conditions. For example, all the microsamples we collected in Ensure shake study (case study #1) were storage at -80 °C in one day, and all the microsamples in the 24/7 study (case study #2) were stored on dry ice immediately, and at -80 °C within one day. We have updated our manuscript to make it clearer for the readers (**page 20, lines 20-21**). Please note that for general use we expect all samples to arrive within 24 hours after collection using overnight services.

Comment #6. I would recommend focusing on a single case study that provides much more information on the rationale for the investigation, the biological significance of the results, and the experimental conditions and their validation (cf international guidelines, as well as data from the QC samples).

Response: We thank the reviewer for the comment. As we mentioned above and in the comment to reviewer #2, the main aim of this study is to develop the multi-omic microsample approach. We included two case studies to highlight the multiple capabilities of microsampling. The Ensure shake study (case study #1) demonstrates using microsampling to measure dynamic response to a specific dietary perturbation, whereas

the 24/7 study (case study #2) demonstrates the use of microsampling to identify correlative relationships between physiological parameters such as heart rate with molecular changes. All the findings in our study should be validated in the new cohort in the future for sure. As a proof of concept study, our goal is to demonstrate new technological capabilities, so the very deep biological interpretation is not the main aim of this study. Thus, we feel the inclusion of both studies makes this a very strong paper.

Additionally, according to the reviewer's comments, we have provided more biological interpretation in the 24/7 study (can be found in response to **Reviewer #3 Comment #1**). We also tried to make the study more readable and thus accessible to the general reader. All the results have been added to our revised manuscript (**page 17, lines 3-14**).